# Neural Implicit Manifold Learning for Topology-Aware Density Estimation

**Brendan Leigh Ross**                                       *brendan@layer6.ai*
*Layer 6 AI*

**Gabriel Loaiza-Ganem**                                     *gabriel@layer6.ai*
*Layer 6 AI*

**Anthony L. Caterini**                                      *anthony@layer6.ai*
*Layer 6 AI*

**Jesse C. Cresswell**                                       *jesse@layer6.ai*
*Layer 6 AI*

**Reviewed on OpenReview:** *https://openreview.net/forum?id=lTOku838Zv*

## Abstract

Natural data observed in $\mathbb{R}^n$ is often constrained to an $m$-dimensional manifold $\mathcal{M}$, where $m < n$. This work focuses on the task of building theoretically principled generative models for such data. Current generative models learn $\mathcal{M}$ by mapping an $m$-dimensional latent variable through a neural network $f_\theta : \mathbb{R}^m \to \mathbb{R}^n$. These procedures, which we call *pushforward models*, incur a straightforward limitation: manifolds cannot in general be represented with a single parameterization, meaning that attempts to do so will incur either computational instability or the inability to learn probability densities within the manifold. To remedy this problem, we propose to model $\mathcal{M}$ as a *neural implicit manifold*: the set of zeros of a neural network. We then learn the probability density within $\mathcal{M}$ with a *constrained energy-based model*, which employs a constrained variant of Langevin dynamics to train and sample from the learned manifold. In experiments on synthetic and natural data, we show that our model can learn manifold-supported distributions with complex topologies more accurately than pushforward models.

## 1 Introduction

Here we undertake the broad statistical task of generative modelling: estimating an unknown probability distribution $P^*$ from a sample $\{x_i\} \subset \mathbb{R}^n$ of datapoints. Real-world distributions are diverse, complex, and often high-dimensional. Despite the apparent difficulty of modelling such data (Cacoullos, 1966), generative modelling methods have been resoundingly successful at synthesizing photorealistic images (Rombach et al., 2022) among other achievements. However, the empirical success of generative modelling has come with unexpected phenomena, such as exploding inverses (Behrmann et al., 2021) and high densities placed on out-of-distribution data (Nalisnick et al., 2019), possibly as a result of its complex geometric (Loaiza-Ganem et al., 2022) or topological (Cornish et al., 2020) structure. These pathologies underscore the necessity of designing mathematically principled generative models whose inductive biases encode reasonable assumptions about the data.

In this work we focus on building a density estimator that correctly specifies the topology of the data manifold. With a thorough analysis of related literature, we highlight that most models for densities on manifolds are *pushforward models*: neural networks $f_\theta : \mathbb{R}^m \to \mathbb{R}^n$ trained to transform an $m$-dimensional prior into a flexible distribution on the data manifold embedded in $\mathbb{R}^n$. As we show, such methods are likely

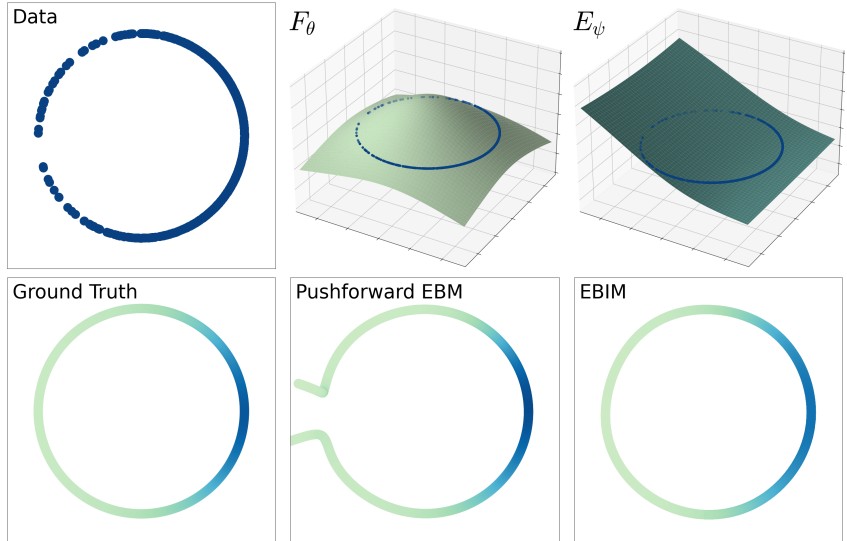

Figure 1: **In the top row**, our EBIM method is depicted on simulated circular data from a von Mises distribution. From left to right: ground truth sample of von Mises data; a manifold-defining function $F_\theta$ learned from the data; and an ambient energy $E_\psi$ trained with constrained Langevin dynamics on the learned manifold. The energy takes the lowest values in areas of high density, precisely as one would expect. **In the bottom row**, manifold learning and density estimation results from the resulting model are juxtaposed with a pushforward baseline. From left to right: the ground truth; a pushforward energy-based model; and an energy-based implicit manifold (ours). By defining the manifold with a constraint $F_\theta(x) = 0$, our method can accurately model data with non-trivial topologies.

to misspecify all but the simplest of data topologies, exposing $f_\theta$ to a frontier of tradeoffs between expressivity and numerical stability (Salmona et al., 2022). This situation is a potential source of the aforementioned pathologies.

To model a much broader class of topologies, we propose a new generative model, the *energy-based implicit manifold* (EBIM). Our approach consists of two steps, which are outlined in Figure 1. Armed with a fundamental theorem from geometry, we first describe a novel way to model data manifolds: *implicitly*, as the zero set of a neural network $F_\theta$. Next, we model the density within the manifold with a *constrained energy-based model*, a likelihood-based model which uses constrained Langevin dynamics to sample points on the learned manifold. We show that EBIMs can be composed with each other akin to standard energy-based models (Hinton, 2002): manifold-defining functions $F_\theta$ along with their energies $E_\psi$ can be combined to take unions and intersections of data manifolds in what we call *manifold arithmetic*. We demonstrate theoretically and empirically that EBIMs can learn manifold-supported densities more accurately than pushforward models. Given its success modelling low- and high-dimensional data alike, EBIMs represent a fruitful direction for mathematically grounded modelling of complex datasets.

## 2 Background, Related Work, and Motivation

### 2.1 Modelling Manifold-Supported Data

**Manifold structure** Commonly, the distribution $P^*$ of interest is supported on a submanifold $\mathcal{M}$ embedded in the ambient space $\mathbb{R}^n$. For example, the manifold hypothesis states that real-world high-dimensional data tends to have low-dimensional submanifold structure (Bengio et al., 2013). Elsewhere, data from engineering or the natural sciences can be manifold-supported due to smooth physical constraints (Mardia et al., 2007; Boomsma et al., 2008; Brehmer & Cranmer, 2020; Cresswell et al., 2022).

We will seek to estimate a probability density on this manifold. Formally, suppose $\{x_i\}$ is a set of samples drawn from probability measure $P^*$ supported on $\mathcal{M}$, an $m$-dimensional Riemannian submanifold of $\mathbb{R}^n$. We focus on the case where $m < n$, so that $\mathcal{M}$ is "infinitely thin" in $\mathbb{R}^n$, meaning $P^*$ does not admit a probability density with respect to the standard Lebesgue measure. However, we may assume it has a density $p^*(x)$ with respect to the Riemannian measure of $\mathcal{M}$. We elaborate formally on this setting in Appendix A.

Models for manifold-supported data have long been of interest in statistics, machine learning, and various applications (Pless & Souvenir, 2009; Diaconis et al., 2013; McInnes et al., 2018). A common theme in machine learning has been to account for – or attribute performance to – underlying manifold structure in data (Ozakin & Gray, 2009; Rifai et al., 2011). In particular, a number of past works have explored Monte Carlo methods on manifolds (Brubaker et al., 2012; Byrne & Girolami, 2013; Zappa et al., 2018), which we put to use here. However, the problem of simultaneously learning a submanifold *and* an underlying density has only become of interest in tandem with recent advances in deep generative modelling (Brehmer & Cranmer, 2020). To our knowledge, all such models fall under the umbrella of *pushforward models*.

**Density estimation with pushforward models**   When manifold-supported, $P^*$ is most commonly modelled as the *pushforward* of some latent distribution:

$$z \sim p_\psi(z), \quad x = f_\theta(z), \tag{1}$$

where $f_\theta : \mathbb{R}^m \to \mathbb{R}^n$ is a smooth mapping given by a neural network and $z \sim p_\psi(z)$ is a (possibly trainable) prior on $m$-dimensional latent space. The resulting model distribution $P_{\theta,\psi}$ is supported on the model manifold[1] $\mathcal{M}_\theta := f_\theta(\mathbb{R}^m)$. This framework encompasses generative adversarial networks (GANs) (Goodfellow et al., 2014; Arjovsky et al., 2017), injective flows (Brehmer & Cranmer, 2020; Caterini et al., 2021), and various regularized autoencoders (Makhzani et al., 2015; Tolstikhin et al., 2018; Ghosh et al., 2020; Kumar et al., 2020). Since we take the support to be an $m$-dimensional submanifold, we rule out models with full-dimensional support, such as bijective normalizing flows (continuous or otherwise) (Rezende & Mohamed, 2015; Dinh et al., 2017; Chen et al., 2018), diffusion models (Sohl-Dickstein et al., 2015; Ho et al., 2020; Song et al., 2021b), and variational autoencoders (VAEs) (Kingma & Welling, 2014; Rezende et al., 2014).

However, not all pushforward models have explicitly computable densities. In recent work, Loaiza-Ganem et al. (2022) outline a general procedure for density estimation with pushforward models, which separates modelling into two components: a *generalized autoencoder*, which embeds the data manifold into $m$-dimensional latent space, and a *density estimator*, which learns the density within the manifold. The generalized autoencoding step treats $f_\theta$ as a decoder, pairing it with a smooth encoder $g_\phi : \mathbb{R}^n \to \mathbb{R}^m$, and trains them to learn $\mathcal{M}$ by mutually inverting each other on the data,[2] such as by minimizing a reconstruction loss $\mathbb{E}_{x \sim P^*} \|x - f_\theta(g_\phi(x))\|^2$. The density estimator $p_\psi(z)$ is then fitted to the encoded data $\{g_\phi(x_i)\}$ via maximum-likelihood. Given a datapoint $x \in \mathcal{M}$, two-step models estimate $p^*(x)$ as follows:

$$p_{\theta,\psi}(x) = p_\psi(z) \left| \det J_{f_\theta}^\top(z) J_{f_\theta}(z) \right|^{-1/2}, \tag{2}$$

where $z := g_\phi(x)$ is the encoding of $x$ and $J_{f_\theta}$ is the Jacobian of $f_\theta$ with respect to its inputs $z$. The fidelity of this estimate depends on the condition $f_\theta(g_\phi(x)) = x$ for all $x \in \mathcal{M}$; in other words, $g_\phi$ must be a right-inverse of $f_\theta$ on $\mathcal{M}$. Injective flow models (Brehmer & Cranmer, 2020; Caterini et al., 2021; Kothari et al., 2021; Ross & Cresswell, 2021) enforce invertibility on $\mathcal{M}_\theta$ with architectural constraints; other two-step models (Xiao et al., 2019; Ghosh et al., 2020; Rombach et al., 2022), like Loaiza-Ganem et al. (2022), achieve this condition at their non-parametric optimum.

**Topological challenges**   Despite the broad applicability of this density estimation procedure, the requisite right-invertibility condition is *effectively impossible* to satisfy for general manifolds $\mathcal{M}$. If $f_\theta(g_\phi(x)) = x$ for all $x \in \mathcal{M}$, then by definition, $g_\phi$ smoothly embeds $\mathcal{M}$ into $\mathbb{R}^m$. This condition presents an immediate topological challenge: $\mathcal{M}$ is an $m$-dimensional manifold, which in general cannot be embedded in $m$-dimensional Euclidean space. In line with the *strong Whitney embedding theorem* (Lee, 2012, pg.135), $\mathcal{M}$ might not be

---

[1] $\mathcal{M}_\theta$ may not formally be a manifold if $f_\theta$ is not an embedding because the resulting image can "self-intersect," but this distinction can be ignored in practice for density estimation models, as we will soon justify.

[2] In particular, $f_\theta$ becomes a left inverse of $g_\phi$ on $\mathcal{M}$, and $g_\phi$ becomes a right inverse of $f_\theta$ on $\mathcal{M}$.

embeddable in Euclidean space of less than $2m$ dimensions.[3] It is thus impossible in the general case for the support of the prior $p_\psi(z)$ to match $\mathcal{M}$ topologically; see the bottom-middle of Figure 1 for an example.

In the presence of this topological mismatch, one might optimistically hope that $\mathcal{M}_\theta$ can sufficiently approximate $\mathcal{M}$ with enough capacity and training. However, Cornish et al. (2020) show that when this is possible, the bi-Lipschitz constant of $f_\theta$ will diverge to infinity, rendering $f_\theta$ either analytically non-injective or numerically unstable, and making density estimates unreliable (Behrmann et al., 2021). Accordingly, the topological woes of pushforward models cannot be "brute-forced" into submission.

Awareness of the data manifold's topology may be necessary for downstream applications such as defending against adversarial examples (Jang et al., 2020) or out-of-distribution detection (Caterini & Loaiza-Ganem, 2021). In the injective normalizing flows literature in particular, there has been interest in learning manifolds with multiple charts (Kalatzis et al., 2021; Sidheekh et al., 2022), which are certainly more expressive than using a single chart. Thus far, such approaches require ancillary models for inference, which can complicate density estimation, and must set the number of charts as a hyperparameter. Multiple charts also may not overlap perfectly, misspecifying the manifold.

## 2.2 Implicitly Defined Manifolds

The aforementioned limitations of pushforward models stem from the inability of smooth embeddings of $\mathbb{R}^m$ to characterize anything but the simplest of manifolds. A richer class of manifolds can be defined *implicitly*, as given by the following fact from differential geometry (Lee, 2012, pg.105):

**The full-rank zero set theorem**    Let $U \subseteq \mathbb{R}^n$ be an open subset of $\mathbb{R}^n$, and let $F : U \to \mathbb{R}^{n-m}$ be a smooth map. If the Jacobian matrix $J_F$ of $F$ has full rank on its zero set $F^{-1}(\{0\}) := \{x \in U : F(x) = 0\}$, then $F^{-1}(\{0\})$ is a properly embedded submanifold of dimension $m$ in $\mathbb{R}^n$.

In this paper, we exploit this theorem by constructing a neural network $F_\theta$ and defining a new model manifold $\mathcal{M}_\theta := F_\theta^{-1}(\{0\})$. We call $F_\theta$ the *manifold-defining function* (MDF) of $\mathcal{M}_\theta$. We refer to such manifolds as *implicitly defined* or *implicit*. These are not to be confused with the unrelated term *implicit generative model*, which has been used to describe both energy-based models (Du & Mordatch, 2019) and some types of pushforward models (Mohamed & Lakshminarayanan, 2016).

The zero sets of neural networks have been employed with great success for one special type of manifold: 3D shapes (Niemeyer & Geiger, 2021). An active subcommunity has formed around learning implicit 3D shapes with varying types of supervision, such as *a priori* shape information (Chen & Zhang, 2019; Mescheder et al., 2019; Park et al., 2019) or 2D images of the object (Niemeyer et al., 2020). For our context, Gropp et al. (2020) propose the most relevant method, which learns a coherent shape from a point cloud without supervision by regularizing gradients. We can reinterpret this as manifold learning, but it can only be applied in the restrictive setting where $m = n-1$. Here we propose a way to fit $F_\theta$ to data manifolds of any dimension $m$ embedded in any dimension $n \geq m$.

## 2.3 Energy-Based Models

Energy-based models (EBMs) have a long history in machine learning (Lecun et al., 2006) and physics (Gibbs, 1902). The energy-based model represents a probability density on $\mathbb{R}^n$ with an energy model $E_\psi : \mathbb{R}^n \to \mathbb{R}$ by way of the relation

$$p_\psi(x) = \frac{e^{-E_\psi(x)}}{\int_{\mathbb{R}^n} e^{-E_\psi(x')}dx'}. \tag{3}$$

EBMs can be trained for maximum likelihood using contrastive divergence (Hinton, 2002). If $P_\psi$ represents the sampling distribution of the EBM with energy $E_\psi$, the likelihood with respect to a single observation $x_i$ has the following gradient:

$$\nabla_\psi \log p_\psi(x_i) = -\nabla_\psi E_\psi(x_i) + \mathbb{E}_{x' \sim P_\psi}\left[\nabla_\psi E_\psi(x')\right]. \tag{4}$$

---

[3]A naive solution would be to increase the model's latent space dimensionality to $2m$; however, this would make the encoded data $\{g_\phi(x_i)\}$ singular in $\mathbb{R}^{2m}$, invalidating density estimates.

The first gradient term minimizes the energy on the true (positive) observations, while the second maximizes the energy on (negative) samples generated by the model over the course of training. As a result, EBMs can be trained by minimizing

$$\mathbb{E}_{x\sim P^*,x'\sim\text{sg}[P_\psi]}\left[E_\psi(x) - E_\psi(x')\right], \tag{5}$$

where $\text{sg}[\cdot]$ denotes the stop gradient operator commonly available in automatic differentiation libraries such as PyTorch (Paszke et al., 2019), meaning that gradients are not propagated through the sampling process during training.

Xie et al. (2016) introduced the first deep EBM for generative modelling. Notably, their method uses Langevin dynamics (Welling & Teh, 2011), a continuous MCMC algorithm, to generate samples from the model $P_\psi$ during both training and test time. One iteration of Langevin dynamics from point $x^{(t)}$ to $x^{(t+1)}$ consists of the following update:

$$x^{(t+1)} = x^{(t)} + \varepsilon r - \frac{\varepsilon^2}{2}\nabla_x E_\psi(x^{(t)}), \tag{6}$$

where $\varepsilon$ is a hyperparameter for the step size and $r \sim N(0, I_n)$ is Gaussian noise. In practice, many iterations of Langevin dynamics samples are necessary to obtain samples close to the stationary distribution, $P_\psi$. Strategies to assist convergence have since become popular in the literature (Nijkamp et al., 2019; Du & Mordatch, 2019; Grathwohl et al., 2020a; Nijkamp et al., 2020).

Some past works incorporate energy-based models with pushforward models. Xiao et al. (2021) model an EBM in the latent space of a VAE, but its training procedure maximizes full-dimensional likelihoods, making it unsuitable for density estimation on manifolds. Che et al. (2020) and Arbel et al. (2021) construct pushforward EBMs by using GAN discriminators to refine the generator's distribution; these models produce distributions on manifolds, but do not admit density estimates. Yoon et al. (2021) propose normalized autoencoders (NAEs), which treat the reconstruction error of an autoencoder as an energy function with the goal of improving its out-of-distribution detection capability, but like Xiao et al. (2021), it is trained as a full-dimensional model, making it unsuitable for manifold-supported density estimates.

# 3 Method

Our method comprises two steps. In the first step (Section 3.1), we implicitly capture the data manifold using a neural network $F_{\theta^*}$. In the second step (Section 3.2), we train a second neural network $E_{\psi^*}$ to capture the data density within the manifold as an energy. Together, these neural networks form a single distributional model, which we call an *energy-based implicit manifold* (EBIM).

## 3.1 Neural Implicit Manifolds

In this section, we describe a practical procedure for modelling data manifolds implicitly using neural networks. Let $F_\theta : \mathbb{R}^n \to \mathbb{R}^{n-m}$ be a smooth neural network with parameters $\theta$; our goal is to optimize it to become a manifold-defining function for $\mathcal{M}$, the data manifold. According to the full-rank zero set theorem of Section 2.2, $F_\theta$ needs to meet three conditions:

1. $F_\theta(x) = 0$ for all $x \in \mathcal{M}$.

2. $F_\theta(x') \neq 0$ for all $x' \notin \mathcal{M}$.

3. $J_{F_\theta}(x)$ has full rank for all $x \in \mathcal{M}$.

Since $\mathcal{M}$ is the support of $P^*$, condition 1 can be encouraged by the constraint $\mathbb{E}_{x\sim P^*}\|F_\theta(x)\| = 0$, which we achieve by minimizing $\mathbb{E}_{x\sim P^*}\|F_\theta(x)\|$.

Condition 2 requires that $F_\theta$ has no zeros outside of the true data manifold. We can satisfy this constraint by locating off-manifold points $x' \in \mathbb{R}^n$ for which $\|F_\theta(x')\|$ is small and maximizing this norm at these points. This goal bears a resemblance to the negative sample term in contrastive divergence, where here $\|F_\theta(\cdot)\|$

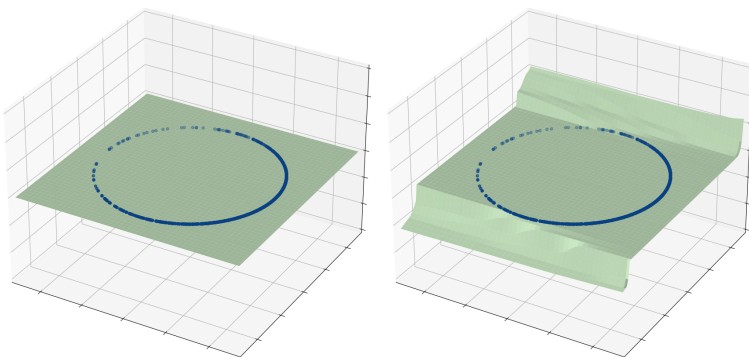

Figure 2: Manifold defining functions $F_\theta$ trained without regularizing negative samples or $J_{F_\theta}$. On the left, a regular neural network, has become completely flat; $F^{-1}(\{0\})$ is the entire space. On the right is the left-inverse of an injective flow, whose Jacobian has full rank analytically, but becomes numerically non-injective. These should be contrasted to the second pane of Figure 1.

acts as an energy function. Taking inspiration from the EBM literature, we thus use Langevin dynamics to sample these negative points from the distribution $P_\theta$ given by the energy $\|F_\theta(\cdot)\|$. Since $\|F_\theta(x)\|$ is a non-negative scalar equal to zero only if $x$ is on the manifold, it is analogous to the reconstruction loss of an autoencoder. Treating it like an energy thus parallels the training procedure of NAEs (Yoon et al., 2021). We emphasize, however, that maximizing $\|F_\theta(x')\|$ on negative samples is a regularizer to satisfy the full-rank zero set theorem which bears a useful resemblance to contrastive divergence. Its purpose is to fit a manifold, not to train an EBM.

Condition 3 is equivalent to ensuring all singular values of $J_{F_\theta}(x)$ are nonzero for $x \in \mathcal{M}$. To achieve this condition, we take inspiration from Kumar et al. (2020). Given their decoder $f_\theta$ and $z \in \mathbb{R}^m$, they make their Jacobian $J_{f_\theta}(z)$ *injective* by bounding $\|J_{f_\theta}(z)u\|$ away from zero for all unit-norm vectors $u \in \mathbb{R}^m$. To implement this bound for all unit-norm $u \in \mathbb{R}^m$ in practice, they sample $u$ uniformly from the unit sphere and minimize the following regularizer with respect to $\theta$:

$$\mathbb{E}_{x \sim P^*, u \sim \mathrm{U}(S^{m-1})} \left[ \left( \eta - \|J_{f_\theta}(z)u\| \right)_+^2 \right], \tag{7}$$

where $\mathrm{U}(S^{m-1})$ is the uniform distribution on the unit sphere $S^{m-1} := \{u \in \mathbb{R}^m : \|u\| = 1\}$, $( \cdot )_+$ denotes the ReLU function, and $\eta$ is a hyperparameter that determines the minimum singular value of $J_{f_\theta}(z)$. We can do the same, except by bounding $\|v^\top J_{F_\theta}(x)\|$ away from zero for all unit-norm $v \in \mathbb{R}^{n-m}$, since we seek to make $J_{F_\theta}(x)$ *surjective*.[4] This serves a similar purpose to the 3D shape learning objective of Gropp et al. (2020), which bounds the $L_2$ norm of the gradient $J_{F_\theta}(x)$ away from zero. However, their process does not readily generalize to higher dimensionalities.

Combining these techniques, we propose the following loss:

$$\mathbb{E}_{x \sim P^*, x' \sim \mathrm{sg}[P_\theta], v \sim \mathrm{U}(S^{n-m-1})} \left[ \|F_\theta(x)\| - \alpha \|F_\theta(x')\| + \beta \left( \eta - \|v^\top J_{F_\theta}(x)\| \right)_+^2 \right], \tag{8}$$

where here $\mathrm{U}(S^{n-m-1})$ is the uniform distribution on the unit sphere in $(n-m)$-dimensional space $S^{n-m} := \{v \in \mathbb{R}^{n-m} : \|v\| = 1\}$ and $\alpha$, $\beta$, and $\eta$ are hyperparameters determining the negative sample weighting, the rank-regularization weighting, and the minimum singular value of $J_{F_\theta}$, respectively. The three terms of this loss respectively encourage conditions 1, 2, and 3 described above. In high-dimensional settings, we find it useful to constrain the *maximum* singular value to $\eta$ as well; this is implemented by replacing the ReLU $( \cdot )_+$ with the identity function. Otherwise, the MDF has a propensity to become very steep during training, destabilizing the negative sample generation process and the resulting gradients of the model weights.

Empirically speaking, the two regularization terms obviate degeneracy in the MDF. Without regularization, $F_\theta$ will converge towards the zero-function, even if we enforce analytical surjectivity in the Jacobian by

---

[4]Note we are here referring to a matrix as injective (resp. surjective) if it has full column (resp. row) rank.

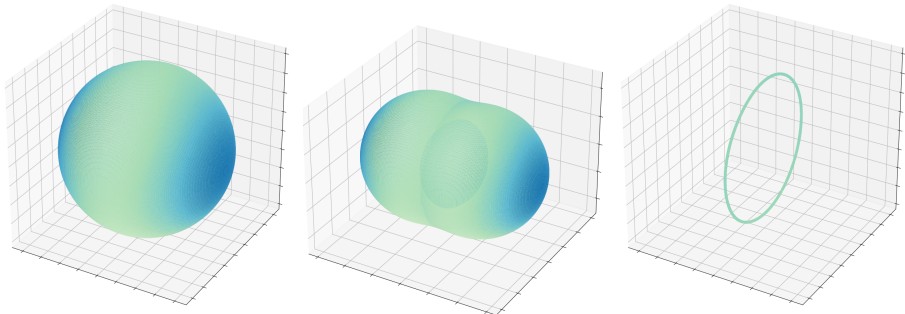

Figure 3: Manifold arithmetic with an implicitly learned sphere. From left to right: a spherical distribution learned by an EBIM; the union of two copies of the same model translated in different directions; and the intersection of the same two copies.

structuring $F_\theta$ as the left-inverse of an injective flow (Kothari et al., 2021) (Figure 2, right). The unregularized flow sends $\mathbb{E}_{x \sim P^*} \|F_\theta(x)\| \to 0$ by bringing its singular values arbitrarily close to zero without learning the manifold. It effectively becomes numerically non-injective, akin to observed instabilities in bijective flows (Cornish et al., 2020; Behrmann et al., 2021).

**Expressivity**  Making the simplifying assumption that neural networks can embody any smooth function (Hornik et al., 1989; Csáji, 2001), we may compare the expressivity of neural implicit manifolds with pushforward manifolds. Pushforward models can model densities on precisely those manifolds which are homeomorphic to a subset of $\mathbb{R}^m$.

On the other hand, a broader class of manifolds can be modelled implicitly. $\mathcal{M}$ can be represented implicitly if and only if it satisfies the technical condition that its normal bundle is "trivial" (Lee, 2012, pg.271). Non-trivial normal bundles are not commonly seen in low-dimensional examples except in non-orientable manifolds such as the Möbius strip or Klein bottle. Though it is unclear whether the manifolds of most natural datasets have trivial normal bundles (e.g. Carlsson et al. (2008) find a dataset of image patches to have the topology of a Klein bottle), it is certainly a broader class than pushforward models can capture.

**Manifold arithmetic**  Some datasets might satisfy multiple constraints, which one might want to learn separately before combining into a mixture or product of models. Since implicit manifold learning can be interpreted as learning a set of constraints, neural implicit manifolds exhibit composability similar to energy-based models (Hinton, 2002; Mnih & Hinton, 2005). If $F_1$ and $F_2$ are MDFs for $\mathcal{M}_1$ and $\mathcal{M}_2$ respectively, then the union $\mathcal{M}_1 \cup \mathcal{M}_2$ is the zero set of the product of functions $x \mapsto F_1(x)F_2(x)$. Concatenating outputs into the function $x \mapsto (F_1(x), F_2(x))$ instead produces the intersection $\mathcal{M}_1 \cap \mathcal{M}_2$. We note that $\mathcal{M}_1 \cup \mathcal{M}_2$ and $\mathcal{M}_1 \cap \mathcal{M}_2$ need not be manifolds anymore, meaning we can combine MDFs to form complex structures that cannot be described with a single manifold (Figure 3). Taking intersections and unions could, for example, be used to model conjunctions or disjunctions of data labelled with multiple overlapping attributes (Du et al., 2020), or to model manifolds with components of different dimensionalities (Brown et al., 2023).

### 3.2 Constrained Energy-Based Modelling

Suppose we have used the techniques from the last section to come up with a fully trained neural implicit manifold $\mathcal{M}_{\theta^*}$, which is henceforth fixed. In this section we introduce the second step in our procedure, the *constrained energy-based model* (CEBM), for density estimation on $\mathcal{M}_{\theta^*}$. Let $E_\psi : \mathbb{R}^n \to \mathbb{R}$ be an energy function represented by a neural network and define the corresponding density as follows:

$$p_{\theta^*, \psi}(x) = \frac{e^{-E_\psi(x)}}{\int_{\mathcal{M}_{\theta^*}} e^{-E_\psi(x')} d\mu(x')}, \quad x \in \mathcal{M}_{\theta^*}, \tag{9}$$

where $d\mu$ can be equivalently thought of as the Riemannian volume form or Riemannian measure of $\mathcal{M}_{\theta^*}$ (see Appendix A for details). Let $P_{\theta^*,\psi}$ be the resulting probability measure (we can think of $P_{\theta^*,\psi}$ as a probability distribution characterized by both the manifold $\mathcal{M}_{\theta^*}$ and the density $p_{\theta^*,\psi}$). Since the energy $E_\psi$ is defined on the full ambient space $\mathbb{R}^n$ but the corresponding model is defined only from its values on $\mathcal{M}_{\theta^*}$, we refer to $p_{\theta^*,\psi}$ as a *constrained energy-based model*.

Having defined $p_{\theta^*,\psi}$ and fixed the manifold $\mathcal{M}_{\theta^*}$, we seek to maximize log-likelihood on the data via gradient-based optimization of $E_\psi$. Since the denominator $\int_{\mathcal{M}_{\theta^*}} e^{-E_\psi(x')} d\mu(x')$ is an intractable integral, we resort to contrastive divergence:

$$\nabla_\psi \log p_{\theta^*,\psi}(x_i) = -\nabla_\psi E_\psi(x_i) + \mathbb{E}_{x' \sim P_{\theta^*,\psi}}[\nabla_\psi E_\psi(x')]. \tag{10}$$

Importantly, the right-most term in Equation 10 is an expectation taken over $P_{\theta^*,\psi}$, so samples from the model are required for optimization.

**Constrained Langevin Monte Carlo** How can one sample from $P_{\theta^*,\psi}$? Du & Mordatch (2019) use Langevin dynamics, a continuous MCMC method, to sample from deep EBMs. For constrained EBMs, standard Langevin dynamics is insufficient, as it will produce off-manifold samples from the energy. We need a manifold-aware MCMC method.

One such method is constrained Hamiltonian Monte Carlo (CHMC), a family of Markov chain Monte Carlo models for implicitly defined manifolds proposed by Brubaker et al. (2012). Our main contribution in this section, aside from defining constrained EBMs, is to show that CHMC – which is typically applied to analytically known manifolds – can be adapted to manifolds that are defined by neural networks. In particular, we show how to avoid the unstable and sometimes memory-prohibitive operation of explicitly constructing the Jacobian of $F_{\theta^*}$, which features prominently in CHMC.

We focus on the special case of constrained Langevin Monte Carlo (CLMC). Fixing a step size $\varepsilon$ and omitting most parameter subscripts for brevity, one iteration from position $x^{(t)}$ to $x^{(t+1)}$ requires two steps:

1. Sample a momentum $r \sim N(0, I_n)$ conditioned on membership of the tangent space of $\mathcal{M}_{\theta^*}$ at $x^{(t)}$. This can be done by sampling $r' \sim N(0, I_n)$ and projecting onto the null space of $J_{F_{\theta^*}}(x^{(t)})$ (written as $J_F$ for clarity):
$$r := r' - J_F^\top (J_F J_F^\top)^{-1} J_F r'. \tag{11}$$

2. Solve for the new position $x^{(t+1)}$ using a constrained Leapfrog step, which entails solving the following system of equations for $x^{(t+1)}$ and the Lagrange multiplier $\lambda \in \mathbb{R}^{n-m}$:

$$x^{(t+1)} = x^{(t)} + \varepsilon r - \frac{\varepsilon^2}{2} \nabla_x E(x^{(t)}) - \frac{\varepsilon^2}{2} J_F(x^{(t)})^\top \lambda \tag{12}$$

$$F(x^{(t+1)}) = 0. \tag{13}$$

Now we describe how Equations 11 and 12 can be computed without constructing $J_F$. With access to efficient Jacobian-vector product (`jvp`) and vector-Jacobian product (`vjp`) routines, such as those available in `functorch` (He & Zou, 2021), any expression in the form of $J_F w = \text{jvp}(F, w)$ for $w \in \mathbb{R}^n$ or $J_F^\top v = (v^\top J_F)^\top = \text{vjp}(v, J)^\top$ for $v \in \mathbb{R}^{n-m}$ is tractable. Furthermore, the inverse term on the right-hand side of Equation 11 can be computed with inspiration from work in injective flows by Caterini et al. (2021) who overcome a similar expression using the conjugate gradients (`CG`) routine (Nocedal & Wright, 2006; Gardner et al., 2018; Potapczynski et al., 2021) and their *forward-backward auto-differentiation trick*. `CG` allows us to compute expressions of the form $A^{-1}b = \text{CG}(A, b)$, where $A$ is an $(n-m) \times (n-m)$ matrix. In particular, `CG` requires access only to the operation $v \mapsto Av$, not the matrix $A$ itself. In our case, $b = J_F r'$, a Jacobian-vector product, and the operation is $v \mapsto J_F J_F^\top v$, which is again computable as a vector-Jacobian product followed by a Jacobian-vector product. Since $J_F$ is a wide matrix, this operation is most efficiently performed using backward-mode followed by forward-mode auto-differentiation, so our method can be termed the *backward-forward* variant.

Equations 12 and 13 can be combined into a single minimization problem which can be easily optimized by first-order methods like stochastic gradient descent (Robbins & Monro, 1951) or second-order methods like

---

**Algorithm 1** Efficient Constrained Langevin Monte Carlo

---

**Require:** trained manifold-defining function $F_{\theta^*}$, energy $E_\psi$, step size $\varepsilon$, step count $k$, initial point $x^{(0)}$
   $x' \leftarrow x^{(0)}$
   **for** $t = 1, \ldots, k$ **do**
      $r' \sim N(0, I_n)$
      // Project $r'$ to tangent space
      $\mathtt{mvp\_JJT}(\cdot) \leftarrow \mathtt{jvp}(F_{\theta^*}, \mathtt{vjp}(\cdot, F_{\theta^*})^\top)$
      $r \leftarrow r' - \mathtt{vjp}\left(\mathtt{CG}\left(\mathtt{mvp\_JJT}(\cdot),\ \mathtt{jvp}(F_{\theta^*}, r')\right), F_{\theta^*}\right)^\top$
      // Take a constrained Leapfrog step along $r$
      $\lambda^* \leftarrow \arg\min_\lambda \left\| F_{\theta^*}\left(x' + \varepsilon r - \frac{\varepsilon^2}{2}\nabla_x E_\psi(x') - \frac{\varepsilon^2}{2}\mathtt{vjp}\left(\lambda, F_{\theta^*}\right)^\top\right)\right\|$
      $x' \leftarrow x' + \varepsilon r - \frac{\varepsilon^2}{2}\nabla_x E_\psi(x') - \frac{\varepsilon^2}{2}\mathtt{vjp}\left(\lambda^*, F_{\theta^*}\right)^\top$
   **return** $x'$

---

L-BFGS (Byrd et al., 1995):

$$\lambda^* = \arg\min_\lambda \left\| F_{\theta^*}\left(x^{(t)} + \varepsilon r - \frac{\varepsilon^2}{2}\nabla_x E_\psi(x^{(t)}) - \frac{\varepsilon^2}{2}J_{F_{\theta^*}}(x^{(t)})^\top \lambda\right)\right\|. \tag{14}$$

In computationally challenging contexts, we can settle for suboptimal solutions at the cost of introducing bias. Once obtained, $\lambda^*$ can be plugged back into Equation 12 to directly calculate $x^{(t+1)}$.

The two steps described above constitute a single iteration of CLMC. In practice, many iterations are required to obtain a sample resembling $P_{\theta^*, \psi}$ (Algorithm 1). Following Du & Mordatch (2019), we use a sample buffer for 95% of generated samples to assist convergence during training. To obtain completely new samples to initialize the CLMC chain, we sample random noise in ambient space and project them to $\mathcal{M}_{\theta^*}$ by computing $\arg\min_{x'} \|F_{\theta^*}(x')\|^2$.

### 3.3 Method Summary

Here we summarize the procedure to fit our entire model, which we call an *energy-based implicit manifold* (EBIM). We start with a dataset $\{x_i\}$ sampled from $P^*$ and belonging to some ground truth manifold $\mathcal{M}$.

1. We first train a network $F_\theta$ to satisfy the full-rank zero set theorem by optimizing Equation 8. Once fully trained, $F_{\theta^*}$ is an MDF for the neural implicit manifold $\mathcal{M}_{\theta^*}$.

2. We next learn the density within $\mathcal{M}_{\theta^*}$ by fitting a constrained EBM, $E_\psi$. The likelihood of $E_\psi$ can be maximized using the gradient in Equation 10, but computing this gradient requires sampling from $E_\psi$. To do so, we run efficient Constrained Langevin Monte Carlo as outlined in Algorithm 1. Once trained, we refer to the energy as $E_{\psi^*}$.

A trained MDF $F_{\theta^*}$ paired with a trained energy $E_{\psi^*}$ together define the manifold and density of the model, $P_{\theta^*, \psi^*}$. We call this $(F_{\theta^*}, E_{\psi^*})$ pair an EBIM.

## 4 Experiments

In this section we demonstrate the efficacy of EBIMs on a diverse range of topologically non-trivial data. Our code is written in `PyTorch` (Paszke et al., 2019). We use `GPyTorch` (Gardner et al., 2018) for conjugate gradients and the marching cubes algorithm of Yatagawa (2021) to plot 2D implicit manifolds in 3D. We generate synthetic data with `Pyro` (Bingham et al., 2019). Network architectures, hyperparameter settings, and further experimental details can be found in Appendix B.

### 4.1 Modelling Non-Trivial Topologies

The current literature on density estimation for non-trivial topologies assumes the manifold is known beforehand (Gemici et al., 2016; Mathieu & Nickel, 2020; Rezende et al., 2020; De Bortoli et al., 2022). Here

we show that EBIMs are the best choice for such distributions in the absence of *a priori* knowledge of the manifold. We reiterate that all manifolds learned in these experiments are determined only based on samples, without additional knowledge. Quantitative comparisons of density estimates are challenging when manifolds are unknown: likelihood values, the usual way to compare density estimators, are uninformative for different learned manifolds. Any test datapoint that is not within the model manifold would result in a model likelihood of 0 because model densities are strictly constrained to their manifolds. Instead, we grade model densities on the basis of the Wasserstein-1 distance to the ground truth (Table 1), which is a rigorous way to measure the distance between two distributions on possibly non-overlapping submanifolds (Arjovsky et al., 2017).

As discussed in Section 2, there are many pushforward density estimation models that could serve as a basis of comparison. We focus on a simple pushforward EBM consisting of an autoencoder with an EBM in the latent space. We experimented with regularizing the autoencoder by training with a Gaussian VAE objective, but it did not learn the manifold as well as a regular autoencoder (Appendix B, Figure 10). Likewise, one could replace the latent EBM with any density estimator (such as a normalizing flow (Brehmer & Cranmer, 2020) or VAE (Dai & Wipf, 2019)), but this would not affect the learned manifold. The pushforward EBM is thus a sufficient baseline for evaluating manifold-learning ability on low-dimensional examples.

**Von Mises mixture**  In our first experiment, we evaluate density estimation ability on 1000 points sampled from a mixture of two von Mises distributions on circles embedded in 2D. Results for an ordinary EBM, a pushforward EBM, and an EBIM are visible in Figure 4. Of note is the topology of the density learned by the pushforward EBM; it is necessarily connected and appears to be homeomorphic to the real line except at two points of self-intersection. The EBIM, in contrast, captures the manifold even in regions of sparsity. The ordinary EBM is not subject to the *topological* limitations of the pushforward EBM, but still lacks the inductive bias to learn the low intrinsic dimension of the data.

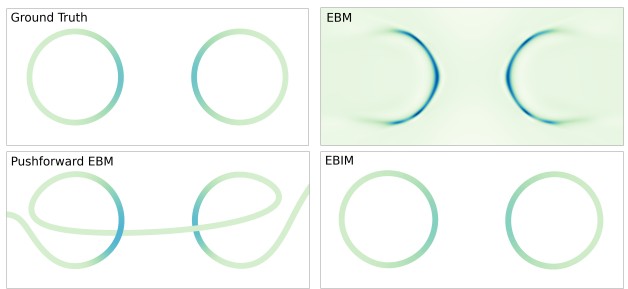

Figure 4: Manifold learning and density estimation results on a balanced, disjoint mixture of two von Mises distributions. Four models are depicted: the ground truth, an ambient EBM, a pushforward EBM, and an EBIM (ours).

**Geospatial data**  Following Mathieu & Nickel (2020), we model a dataset of global flood events from the Dartmouth Flood Observatory (Brakenridge, 2010), embedded on a sphere representing the Earth. Despite the relative sparsity of floods compared to previous datasets (they only occur on land), the EBIM still perfectly learns the spherical shape of the Earth (Figure 5). The pushforward EBM represents the densities fairly well, but struggles to learn the sphere and places some density off of the true manifold. Note that the EBIMs and pushforward EBMs are plotted using a triangular mesh and mesh grid, respectively, due to the difference in how they are defined.

**Amino acid modelling**  The structure of some amino acids can be characterized by a pair of dihedral angles and thus possesses toroidal geometry. Designing flexible probabilistic models for torus-supported data is consequently of interest in the bioinformatics literature on protein structure prediction (Singh et al., 2002; Mardia et al., 2007; Ameijeiras-Alonso & Ley, 2020), and so amino acid angle data is a practical candidate for evaluating the density estimation ability of EBIMs. In Figure 6, we compare an EBIM with a pushforward EBM using an open-source amino acid dataset available from the `NumPyro` software package (Phan et al., 2019). Remarkably, our manifold-defining function learns the torus well in the presence of sparse data. We postulate this is because the torus is the simplest manifold matching the data's curvature. On the other hand, the pushforward EBM was unable to reliably model the manifolds. This stark drop in performance is concerning because one might reasonably expect higher-dimensional datasets to have more complex topologies than a simple torus, but the corresponding misbehaviour of the pushforward model would be impossible to visualize and difficult to detect.

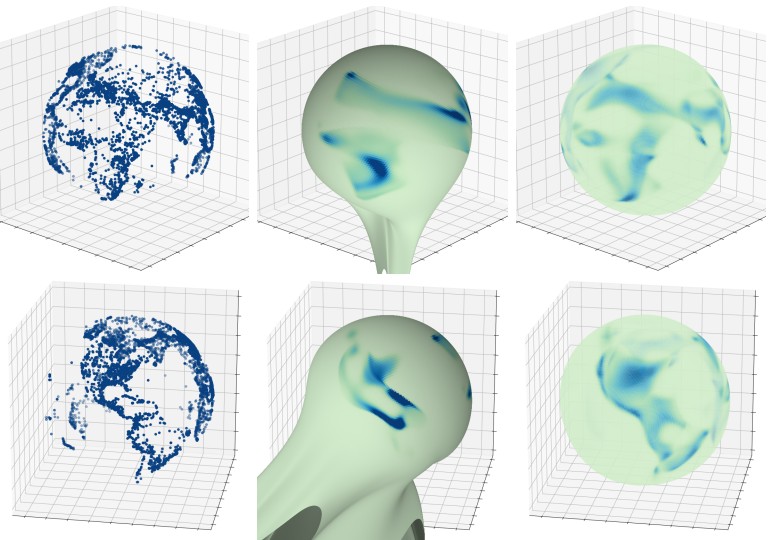

Figure 5: Manifold learning and density estimation results on flood location data. From left to right with two different viewpoints (top and bottom): the ground truth data; a pushforward EBM; and an EBIM (ours).

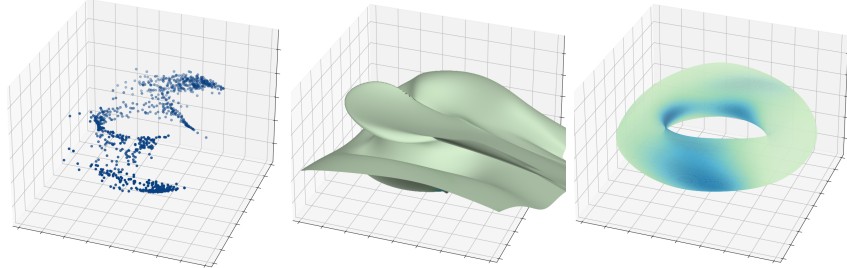

Figure 6: Manifold learning and density estimation results on the glycine angle data. From left to right: the ground truth data; a pushforward EBM; and an EBIM (ours).

Table 1: Mean Wasserstein distances with standard errors over 3 runs (lower is better).

| Dataset | PEBM | EBIM (ours) |
|---|---|---|
| Von Mises Mixture | $0.013 \pm 0.001$ | $\mathbf{0.002 \pm 0.001}$ |
| Geospatial | $0.056 \pm 0.017$ | $\mathbf{0.014 \pm 0.003}$ |
| Amino Acid Modelling | $0.042 \pm 0.013$ | $\mathbf{0.026 \pm 0.001}$ |

Numerical results for the above three topologically interesting datasets are given in Table 1. In all cases the EBIM more closely replicates the data distribution than the pushforward EBM, and usually with less variation over runs.

## 4.2 Modelling Image Manifolds

**Image generation** In this section we show that EBIMs can be scaled to higher-dimensional data manifolds: MNIST (LeCun et al., 1998) and Fashion MNIST (Xiao et al., 2017). For the manifold dimension we select 16 *a priori*; this value is close to intrinsic dimension estimates of MNIST and Fashion MNIST in the literature (Pope et al., 2021; Zheng et al., 2022; Brown et al., 2023). We provide three baseline comparisons: a vanilla autoencoder with an EBM in the latent space (AE+EBM), a variational autoencoder with an EBM in the latent space (VAE+EBM), and a normalized autoencoder (NAE) (Yoon et al., 2021). In short, NAEs are EBMs whose energies are defined by the reconstruction loss of an autoencoder and which use its latent

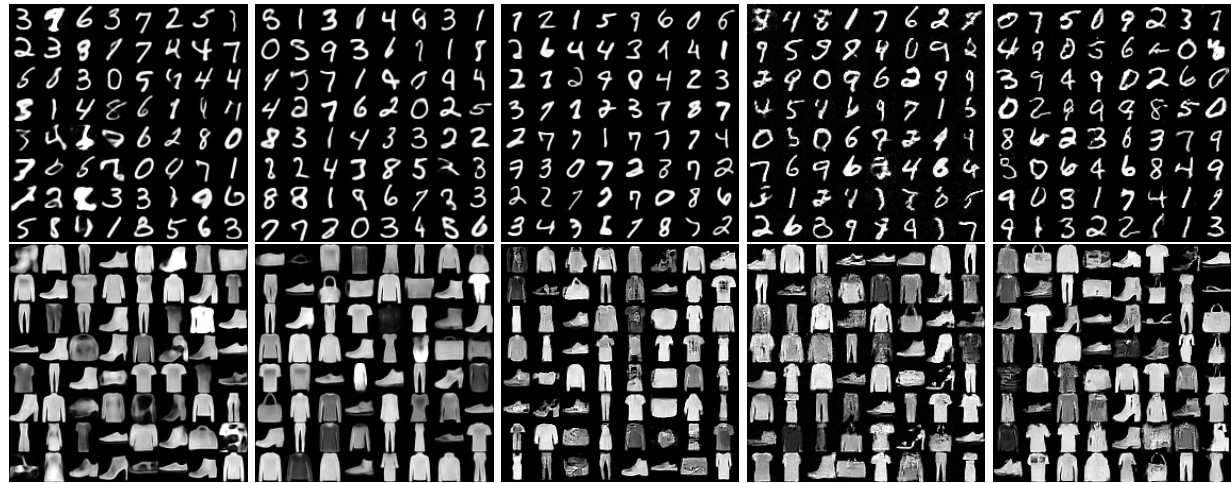

Figure 7: MNIST (top) and Fashion MNIST (bottom) samples. Columns from left to right: AE + EBM, VAE + EBM, NAE, IM (ours, ablation), and EBIM (ours).

Table 2: Mean FID scores with standard errors over 5 runs (lower is better).

| DATASET | AE+EBM | VAE+EBM | NAE | IM (ours, ablation) | EBIM (ours) |
|---------|--------|---------|-----|---------------------|-------------|
| MNIST | $19.8 \pm 0.4$ | $17.9 \pm 0.6$ | $17.0 \pm 0.3$ | $19.2 \pm 0.5$ | $\mathbf{15.3 \pm 1.0}$ |
| FMNIST | $52.7 \pm 0.3$ | $54.1 \pm 0.5$ | $\mathbf{28.2 \pm 2.6}$ | $39.3 \pm 0.3$ | $\mathbf{30.7 \pm 0.6}$ |

space to perform efficient *on-manifold initialization* of Langevin dynamics. They were designed for out-of-distribution detection and cannot provide density estimates on manifolds, our stated goal, but we include them for their training similarities to EBIMs. We also report the quality of samples generated from the implicit manifolds (IMs) prior to fitting the density with a CEBM (i.e. we treat $\|F_{\theta^*}(\cdot)\|$ as an energy function in $\mathbb{R}^n$). Samples from all models are provided in Figure 7 with FID scores (Heusel et al., 2017) in Table 2 for reference.[5]

The EBIM outperforms the two pushforward models, the AE+EBM and VAE+EBM, on both datasets, as well as the NAE on MNIST. However, the EBIM slightly underperforms the NAE on Fashion MNIST. One cause is that Langevin dynamics can be run more efficiently with NAEs; it is partly run in low-dimensional latent space, and requires only one network pass per step. In contrast, constrained Langevin Monte Carlo must be run in high-dimensional ambient space and requires many network passes per step (Algorithm 1). As a result, we only run one step per training step for image data, whereas NAEs are trained with 10 latent steps and 50 ambient steps per training step. Even so, NAEs take about 0.5 seconds per training step, while CEBMs take about 3 seconds. Another possible reason for the strength of the NAEs is that they can correct for topological constraints through the NAE's ambient energy function, which allows it to sample off-manifold points. This topological correction also explains its outperformance of the two pushforward models.

The fact that EBIMs generate images of comparable quality suggests that they make up for the inefficiency of constrained Langevin dynamics with other advantages: namely, an improved ability to learn manifold topologies. In particular, the implicit manifold (IM) by itself was able to generate samples of decent quality using Langevin dynamics as described in Section 3.1, whereas samples we produced from training NAEs without on-manifold initialization in latent space (i.e. by just training the manifold without using the density within) were not digit- or clothing-like at all. This result indicates EBIMs may be learning manifolds better, but the sampling procedure within their manifolds is simply less efficient.

---

[5]FID scores measure some distance between two probability distributions, but it is unclear whether they measure the fidelity of a learned manifold and the density within (Stein et al., 2023). We report it as it is the most popular metric, though its suitability for simple greyscale datasets like MNIST and Fashion MNIST is questionable.

Despite its inefficiency, however, constrained Langevin Monte Carlo demonstrates clear value in the fact that EBIMs produce better samples than the original IMs. This suggests that the CEBM step does indeed refine the density of the IM and that this refinement translates measurably into sample quality.

**Manifold arithmetic** In this section, we apply manifold arithmetic to image manifolds. We denote by $\mathcal{M}_i$ the manifold corresponding to class $i$ of MNIST. Here we consider two overlapping image manifolds based on subsets of the MNIST manifold: (1) $\mathcal{M}_1 \cup \mathcal{M}_4$ and (2) $\mathcal{M}_4 \cup \mathcal{M}_7$. In our experiment, we train two MDFs; one fitted to each of $\mathcal{M}_1 \cup \mathcal{M}_4$ and $\mathcal{M}_4 \cup \mathcal{M}_7$. We then take the intersection of the two learned manifolds by concatenating the outputs of the MDFs as outlined in Section 3.1. This new MDF should define the manifold $(\mathcal{M}_1 \cup \mathcal{M}_4) \cap (\mathcal{M}_4 \cup \mathcal{M}_7)$. We sample from this intersection manifold using Langevin dynamics as described in Section 3.1. An uncurated collection of samples is visible in Figure 8. Most samples belong to $\mathcal{M}_4$, as expected, but interestingly, some samples are ambiguous and appear to belong to $\mathcal{M}_1 \cap \mathcal{M}_7$ as well.

Figure 8: Uncurated samples from $(\mathcal{M}_1 \cup \mathcal{M}_4) \cap (\mathcal{M}_4 \cup \mathcal{M}_7)$.

We also tried directly computing the union $(\mathcal{M}_1 \cup \mathcal{M}_4) \cup (\mathcal{M}_4 \cup \mathcal{M}_7)$, but Langevin dynamics was unstable for the product of two MDFs. Instead, to generate images from unions of manifolds, we recommend sampling from a mixture of the two image manifolds: by first sampling from a categorical distribution indicating which model to sample from, and then from the corresponding manifold. Still, the manifold arithmetic approach to unions is unique in providing an implicit representation of the new manifold (eg. for visualization, as in Figure 3).

## 5 Conclusion

In this paper we observed that most existing techniques to jointly learn data manifolds and densities can be described as *pushforward models*. These models must become near-homeomorphisms, an overly strong topological limitation, in order to provide reliable density estimates. To circumvent this limitation, we introduced the *energy-based implicit model*, which was outlined in two parts. First, we proposed to learn the data manifold *implicitly* with a neural network $F_\theta$. We then proposed the *constrained EBM*, a new type of EBM for modelling data on neural implicit manifolds. In both cases, we showed how the computation of the Jacobian of $F_\theta$ can be "tamed" using stochastic estimates and automatic differentiation tricks inspired by the injective flows literature (Kumar et al., 2020; Caterini et al., 2021) which frequently grapples with non-square Jacobians. We used these techniques to model distributions with complex topologies; the resulting efficiency gains allowed us to scale the resulting model up to image data.

Although we have covered the limitations of pushforward models when used for density estimation, we highlight here some of their advantages over our model. Primarily, pushforward models come with latent representations of data, which have myriad uses such as explainability and artificial reasoning (Higgins et al., 2017; Mathieu et al., 2019) and efficient density estimation in the latent space. A promising direction for future work is to combine these benefits with those of EBIMs.

Another direction would be to sidestep pushforward models entirely and investigate whether densities on manifolds can be extracted from diffusion models. Diffusion models are known to provide full-dimensional densities once converted into continuous normalizing flows (Song et al., 2021a;b). However, these are not densities on manifolds. On the other hand, Pidstrigach (2022) has shown that, if a diffusion model's score function is allowed to explode towards infinity as the timestep approaches 0, it is capable of modelling manifold-supported data. Being able to extract densities on the manifold under these conditions would be interesting, though to our knowledge, there is no known way to do so.

Future work might also study extra applications of the manifold recovered by $F_\theta$, such as unsupervised out-of-distribution detection. Informally, since $||F_\theta(x)||$ measures distance from the data manifold, one might expect its value to be larger for out-of-distribution points. However, anecdotally, we occasionally observed in-

distribution (Fashion-MNIST) points being assigned larger $||F_\theta(x)||$ values than out-of-distribution (MNIST) points, mirroring the pathologies observed by Nalisnick et al. (2019) for density estimators. This behaviour suggests that our network for $||F_\theta(x)||$ is inhibited in this task by similar inductive biases (Kirichenko et al., 2020). An interesting direction would be to understand this problem in the context of manifold learning.

Our model also inherits all the limitations of training EBMs; for example, it relies on the assumption that Langevin dynamics converges, which occurs only with infinite steps. Sampling remains slower than normal EBMs due to the complexity of constrained Langevin dynamics. Constrained EBMs might thus benefit from training methods that do not involve sampling, such as the Stein discrepancy (Grathwohl et al., 2020b) or score-matching (Hyvärinen, 2005; Song & Kingma, 2021; De Bortoli et al., 2022). This might help the model scale to larger datasets like CIFAR-10 (Krizhevsky & Hinton, 2009), which we were able to train an implicit manifold on, but on which we found tuning of the CEBM intractable.

### Acknowledgments

We thank Emile Mathieu for assisting us with the global flood event dataset. We also thank Kin Kwan Leung for our discussions about which submanifolds can be modelled implicitly.

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

# A  Formal setting

Here we expand on the formal setting in which we seek to perform density estimation.

**Geometry**  Let $\mathcal{M}$ be an $m$-dimensional orientable[6] Riemannian submanifold of ambient space $\mathbb{R}^n$ where $m < n$. Formally this refers to the pair $(\mathcal{M}, \mathbf{g})$, where $\mathcal{M} \subseteq \mathbb{R}^n$ is a manifold and $\mathbf{g}$ is the Riemannian metric inherited from ambient Euclidean space. In other words, $\mathbf{g}$ is the restriction of the canonical Euclidean metric, which is characterized by the standard dot product between vectors, to vectors which are tangent to $\mathcal{M}$. The metric $\mathbf{g}$, which is typically implied, captures the curvature information we would like to associate with $\mathcal{M}$.

A manifold's Riemannian metric gives rise to a unique differential form known as the Riemannian volume form $d\mu$, which allows for the integration of continuous, compactly supported, real-valued functions $h$ over the Riemannian manifold (Lee, 2012):

$$\int_{\mathcal{M}} h \ d\mu. \tag{15}$$

**Probability**  Let $\{x_i\}$ be observed samples drawn from $P^*$, a probability measure supported on $\mathcal{M}$. Since $\mathcal{M}$ has a lower intrinsic dimension than $\mathbb{R}^n$, it is "infinitely thin." In other words, $P^*(\mathcal{M}) = 1$ while the (Lebesgue) volume of $\mathcal{M}$ is 0, meaning no probability density integrated over the ambient space can be used to represent $P^*$. Formally stated, $P^*$ is not absolutely continuous with respect to the Lebesgue measure on $\mathbb{R}^n$.

Instead, we require a new way to define the volumes of subsets of $\mathcal{M}$. We can then formally define a probability density $p^*$ over $\mathcal{M}$ and integrate with respect to this volume to obtain probabilities. The volume form $d\mu$ on $\mathcal{M}$ is the answer; the probability of a set $S \subseteq \mathcal{M}$ can be computed as follows:

$$P^*(S) = \int_S p^*(x) \ d\mu(x). \tag{16}$$

We note that the volume form $d\mu$ from differential geometry is not technically a measure in the sense of measure theory. This obstacle is minor: $d\mu$ can be extended to a true measure by a common measure-theoretic tool known as the Riesz-Markov-Kakutani representation theorem[7] (Rudin, 1987). Thus we may identify $d\mu$ with a measure $\mu$ on $\mathcal{M}$ which produces volumes of Borel sets in $\mathcal{M}$ and which we call the Riemannian measure of $\mathcal{M}$ (Pennec, 1999).

Formally, we require $P^*$ to be absolutely continuous with respect to $\mu$, and we thus write that $p^*$ is the Radon-Nikodym derivative of $P^*$ with respect to $\mu$: $p^* = \frac{dP^*}{d\mu}$. This is the ground-truth density function we seek to model in this work.

# B  Experimental Details

For all experiments, we use feedforward networks with SiLU activations (Hendrycks & Gimpel, 2016; Ramachandran et al., 2017). All models are trained with the Adam optimizer (Kingma & Ba, 2015) with the default PyTorch parameters, except for the learning rate which is set as described below.

## B.1  Low-Dimensional Data

All EBMs, EBIMs, and pushforward EBMs in this subsection are trained with a buffer size of 1000, from which we initialize each Langevin dynamics sample with 95% probability. We do not use spectral normalization for EBMs: we found it harmed the quality of density estimates. Initial noise for the EBIM is sampled

---

[6]We focus on orientable manifolds because, as discussed in Section 3.1, manifolds with non-trivial normal bundles, which include non-orientable manifolds, cannot be modelled implicitly. This formal problem setting can be expanded to non-orientable submanifolds too, but requires a slightly different construction than is used in this section.

[7]In the reference and sometimes in general, this theorem is called the Riesz representation theorem, which can also refer to a different theorem about Hilbert spaces.

Table 3: Statistics of estimated dataset distances to the manifold.

| Experiment | EBIM | | | | Pushforward EBM | | | |
|---|---|---|---|---|---|---|---|---|
| | Min | Median | Mean | Max | Min | Median | Mean | Max |
| Motivating example | $0.059 \times 10^{-5}$ | $0.30 \times 10^{-2}$ | $0.34 \times 10^{-2}$ | 0.012 | $0.376 \times 10^{-5}$ | $0.27 \times 10^{-2}$ | $0.33 \times 10^{-2}$ | 0.153 |
| Manifold arithmetic | $2.387 \times 10^{-5}$ | $0.77 \times 10^{-2}$ | $0.79 \times 10^{-2}$ | 0.018 | - | - | - | - |
| Von Mises mixture | $0.006 \times 10^{-5}$ | $0.73 \times 10^{-2}$ | $0.98 \times 10^{-2}$ | 0.045 | $0.810 \times 10^{-5}$ | $0.47 \times 10^{-2}$ | $0.51 \times 10^{-2}$ | 0.095 |
| Geospatial data | $0.132 \times 10^{-5}$ | $0.24 \times 10^{-2}$ | $0.24 \times 10^{-2}$ | 0.009 | $20 \times 10^{-5}$ | $0.16 \times 10^{-2}$ | $0.17 \times 10^{-2}$ | 0.016 |
| Amino acid modelling | $9.176 \times 10^{-5}$ | $2.80 \times 10^{-2}$ | $2.86 \times 10^{-2}$ | 0.16 | $6.96 \times 10^{-5}$ | $0.92 \times 10^{-2}$ | $1.72 \times 10^{-2}$ | 0.271 |

uniformly from a box in ambient space containing the ground truth manifold and then projected to the manifold by solving for $\arg\min_{x'} \|F_{\theta^*}(x')\|^2$ with L-BFGS using strong Wolfe line search. Equation 14 is also optimized using a single step of L-BFGS with strong Wolfe line search. All models were tuned by hand for visual performance. Training times are reported below, but we caution that models were not tuned for runtime, so the raw times should not be compared between models to evaluate efficiency. In general the constrained EBM is the slowest, followed by the MDF and ordinary EBMs, then the two stages of the pushforward EBM.

To plot the EBIM densities, we estimate the normalizing constants using Monte Carlo. Since the learned MDFs always provide very good approximations of the true manifolds, we estimate each normalizing constant using uniform samples from the *ground truth* manifold for convenience. To plot the pushforward EBM densities, we estimate the normalizing constants in latent space with Monte Carlo estimates based on uniform sampling within the clamped bounds. We then compute pushforward densities with Equation 2.

All low-dimensional experiments were performed on an Intel Xeon Silver 4114 CPU.

To evaluate Wasserstein-1 distances, we discretize the space into cells (with a granularity of $100 \times 100$ for the 2D example and $30 \times 30 \times 30$ for the 3D examples) and use a large quantity of samples from the model manifold to determine whether the model manifold exists in each cell. We then use these samples to evaluate the average density value per cell. These density values are normalized over all cells and used to compute a discrete Wasserstein-1 distance with the `pot` library (Flamary et al., 2021), where distances between pairs of cells are encoded as the distance between their centres.

We provide additional quantitative results in Table 3. Here we estimate the distance of each training point to the manifold using an optimization procedure, and report minimum, median, mean, and maximum distances over the training set. Nearest-point estimates must be computed differently for EBIMs and pushforward EBMs, and therefore estimates for each model are prone to different sources of error, so these metrics should be used only with caution as a basis for comparison. For the EBIM with MDF $F_{\theta^*}$, we compute the nearest point on the manifold to datapoint $x_i$ as

$$x^* = \arg\min_{x'} \|x' - x_i\|^2 + 10^{10}\|F_{\theta^*}(x')\|^2,$$

where $x'$ has been initialized to $x_i$. For the pushforward EBM with encoder-decoder pair $(f_{\theta^*}, g_{\phi^*})$, we compute the nearest point as $f_{\theta^*}(z^*)$, where

$$z^* = \arg\min_{z} \|f_{\theta^*}(z) - x_i\|^2,$$

where $z$ has been initialized to $g_{\phi^*}(x_i)$.

**Motivating example (Figure 1)**   We sampled 1000 points from a von Mises distribution on a unit circle centred at $(0, 0)$ with the mode located at $(1, 0)$ and a concentration of 2.

The MDF for the EBIM consisted of 3 hidden layers with 8 units per hidden layer. The MDF was trained for 300 epochs with a batch size of 50, a learning rate of 0.01, $\eta = 1$, $\alpha = 0.3$, and $\beta = 10$. Langevin dynamics with run with $\varepsilon = 0.1$ and a step size of 10. Training took 4 minutes, 8 seconds.

The energy function for the EBIM consisted of 2 hidden layers with 32 units per hidden layer. It was trained for 20 epochs with a batch size of 50, a learning rate of 0.01, gradients clipped to a norm of 1, and energy

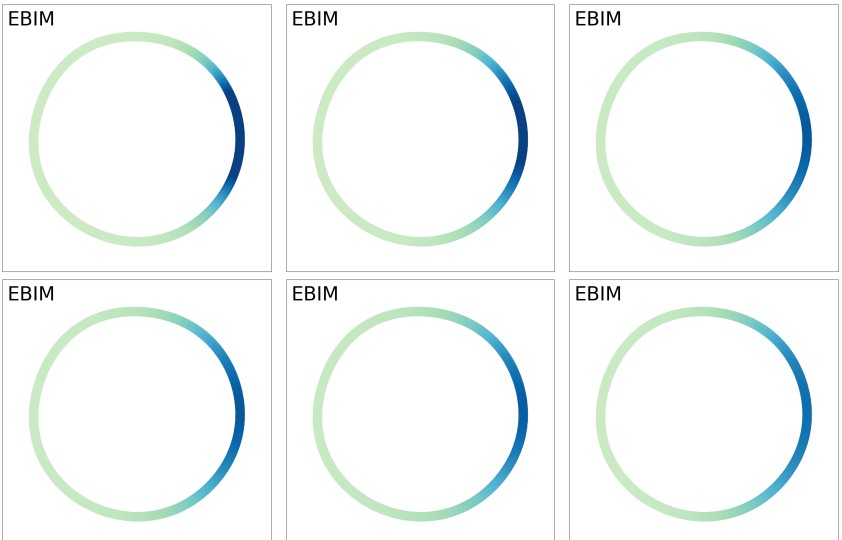

Figure 9: EBIM manifold learning and density estimation results on a von Mises distribution where Langevin dynamics during training has been run (with no replay buffer) with different step counts. From left to right and top to bottom, step counts per training step: 1, 3, 5, 10, 20, 40. The setting of 20 langevin dynamics steps is sufficient for convergence.

magnitudes regularized with a coefficient of 0.1. Langevin dynamics at each training step were run for 10 steps with $\varepsilon = 0.3$, a step size of 1, and energy gradients clamped to maximum values of 0.1 at each step. Training took 2 minutes, 51 seconds. In Figure 9, we evaluate the effect of the Langevin dynamics step count on training dynamics, where we vary the step size (and remove the training buffer, as this effectively increases the average step count). Fewer steps leads to a more peaked mode because the estimated model distribution is overly smooth when estimating the right-hand side of Equation 10.

The pushforward EBM's encoder and decoder each had 3 hidden layers with 32 units per hidden layer. They were jointly trained for 300 epochs with a batch size of 50, a learning rate of 0.001, and gradients clipped to a norm of 1. Training took 31.6 seconds.

The pushforward EBM's energy function had 3 hidden layers and 32 units per hidden layer. It was trained for 200 epochs with a batch size of 50, a learning rate of 0.01, gradients clipped to a norm of 1, and energy magnitudes regularized with a coefficient of 0.1. Langevin dynamics at each training step were run for 20 steps with $\varepsilon = 0.5$, a step size of 10, and energy gradients clamped to maximum values of 0.03 at each step. Training took 3 minutes, 5 seconds.

**Manifold arithmetic** Figure 3 depicts two modes of composition for EBIMs. The EBIM depicted on the left is learned from 1000 points sampled from a balanced mixture of two projected normal distributions. Each component was a normal distribution with unit diagonal covariance centred at $(1, 0, 0)$ and $(-1, 0, 0)$, respectively, before being projected to the sphere. After this, with no additional training, we manipulate it to create new probability models. First, two copies of the learned model are translated by 0.5 units in opposite directions.

- A new model given by the union of these two copies is depicted in the middle pane of Figure 3: it consists of the product of their MDFs and a balanced mixture of their corresponding energies. Note that the new surface self-intersects, and is no longer formally an embedded submanifold.

- Another new model given by the intersection of these two copies is visible in the final pane. By concatenating the output of the MDFs and summing the corresponding energies, we arrive at a circle embedded in three dimensions.

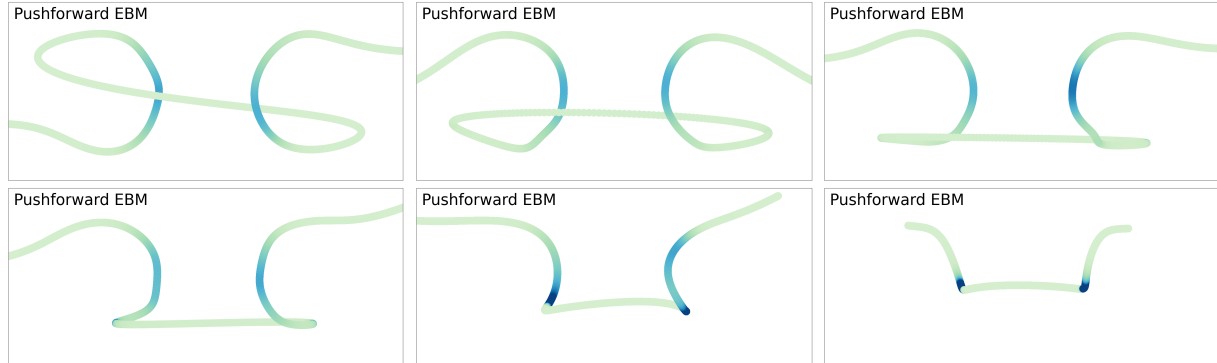

Figure 10: Manifold learning and density estimation performance for different weightings $\beta$ on the KL-divergence term of the VAE loss. From left to right, top to bottom: $\beta = 0.01, \beta = 0.03, \beta = 0.05, \beta = 0.1, \beta = 0.2, \beta = 0.4$.

The MDF for the EBIM consisted of 3 hidden layers with 8 units per hidden layer. The MDF was trained for 300 epochs with a batch size of 100, a learning rate of 0.01, $\eta = 0.1$, $\alpha = 0.3$, and $\beta = 10$. Langevin dynamics at each training step were run for 10 steps with $\varepsilon = 0.1$, a step size of $\varepsilon^2$, and energy gradients clamped to maximum values of 0.03 at each step. Training took 2 minutes 11 seconds.

The energy function for the EBIM consisted of 2 hidden layers with 32 units per hidden layer. It was trained for 20 epochs. We used a batch size of 100, a learning rate of 0.01, gradients clipped to a norm of 1, and energy magnitudes regularized with a coefficient of 1. Langevin dynamics at each training step was run for 10 steps with $\varepsilon = 0.3$, a step size of $\varepsilon^2$, and energy gradients clamped to maximum values of 0.03 at each step. Training took 2 minutes, 15 seconds.

**Von Mises mixture**  We sampled 1000 points from a balanced mixture of two von Mises distributions with concentration 2 on circles of unit radius. Respectively, they are centred at $(-2, 0)$ and $(2, 0)$ with modes at $(-1, 0)$ and $(1, 0)$ (or, at polar angles of 0 and $\pi$ with respect to the centre of each circle).

The MDF for the EBIM consisted of 3 hidden layers with 8 units per hidden layer. The MDF was trained for 500 epochs with a batch size of 100, a learning rate of 0.01, $\eta = 1$, $\alpha = 0.3$, and $\beta = 1$. Langevin dynamics at each training step was run for 20 steps with $\varepsilon = 0.1$, a step size of 10, and energy gradients clamped to maximum values of 0.03 at each step. Training took 3 minutes, 47 seconds.

The energy function for the EBIM consisted of 3 hidden layers with 32 units per hidden layer. It was trained for 10 epochs with a batch size of 100, a learning rate of 0.01, gradients clipped to a norm of 1, and energy magnitudes regularized with a coefficient of 0.3. Langevin dynamics at each training step were run for 10 steps with $\varepsilon = 0.5$, a step size of 1, and energy gradients clamped to maximum values of 0.1 at each step. Training took 1 minute, 23 seconds.

The (ambient) EBM consisted of 3 hidden layers with 32 units per hidden layer. It was trained for 200 epochs with a step size of 10. We used a batch size of 100, a learning rate of 0.01, gradients clipped to a norm of 1, and energy magnitudes regularized with a coefficient of 0.5. Langevin dynamics at each training step were run for 10 steps with $\varepsilon = 0.1$ and energy gradients clamped to maximum values of 0.03 at each step. Training took 1 minute, 14 seconds at each step.

The pushforward EBM's encoder and decoder each had 3 hidden layers with 32 units per hidden layer. It was trained for 300 epochs with a batch size of 100, a learning rate of 0.001, and gradients clipped to a norm of 1. Training took 34.1 seconds. We also tried training the autoencoder using a variational autoencoder loss, but found that to learn the manifold properly, the KL term had to be heavily downweighted near the point of nonexistence. In Figure 10 we show how manifold learning ability deteriorates as the KL-weighting is increased.

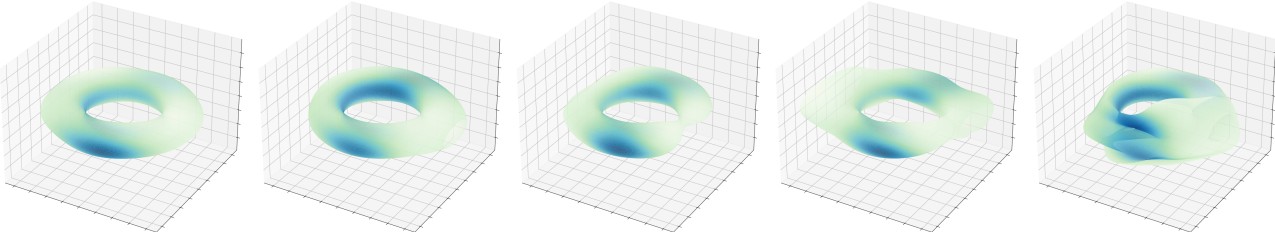

Figure 11: EBIM manifold learning and density estimation results on the glycine angle data for different values of $\eta$, the hyperparameter setting the boundary under which singular values will be penalized by the Jacobian regularization term. From left to right: $\eta = 0.3$, $\eta = 1$, $\eta = 2$, $\eta = 3$, and $\eta = 5$.

The pushforward EBM's energy function had 3 hidden layers and 32 units per hidden layer. It was trained for 300 epochs with a batch size of 100, a learning rate of 0.01, gradients clipped to a norm of 1, and energy magnitudes regularized with a coefficient of 0.1. Langevin dynamics at each training step was run for 20 steps with $\varepsilon = 1.0$, a step size of 10, and energy gradients clamped to maximum values of 0.03 at each step. Training took 2 minutes, 21 seconds.

**Geospatial data**  We modelled floods from the Dartmouth Flood Observatory's global active archive, which is available without charge for research and education purposes.

The MDF for the EBIM consisted of 3 hidden layers with 8 units per hidden layer. The MDF was trained for 300 epochs with a batch size of 50, a learning rate of 0.01, $\eta = 1$, $\alpha = 0.3$, and $\beta = 10$. Langevin dynamics at each training step was run for 20 steps with $\varepsilon = 0.1$, a step size of 10, and energy gradients clamped to maximum values of 0.03 at each step. Training took 19 minutes, 43 seconds.

The energy function for the EBIM consisted of 4 hidden layers with 32 units per hidden layer. It was trained for 100 epochs. We used a batch size of 50, a learning rate of 0.01, gradients clipped to a norm of 1, and energy magnitudes regularized with a coefficient of 1. Langevin dynamics at each training step was run for 5 steps with $\varepsilon = 0.1$, a step size of $\varepsilon^2$, and energy gradients clamped to maximum values of 0.03 at each step. Training took 1 hour, 52 seconds.

The pushforward EBM's encoder and decoder each had 4 hidden layers with 32 units per hidden layer. They were jointly trained for 500 epochs with a batch size of 100, a learning rate of 0.001, and gradients clipped to a norm of 1. Training took 2 minutes, 33 seconds.

The pushforward EBM's energy function had 4 hidden layers and 32 units per hidden layer. It was trained for 50 epochs with a batch size of 50, a learning rate of 0.01, gradients clipped to a norm of 1, and energy magnitudes regularized with a coefficient of 0.1. Langevin dynamics at each training step were run for 60 steps with $\varepsilon = 0.5$, a step size of 10, and energy gradients clamped to maximum values of 0.03 at each step. Training took 9 minutes, 9 seconds.

**Amino acid modelling**  The MDF for the EBIM consisted of 2 hidden layers with 8 units per hidden layer. The MDF was trained for 500 epochs with a batch size of 50, a learning rate of 0.01, $\eta = 0.3$, $\alpha = 0$, and $\beta = 1$. We found that increasing $\eta$, the smallest singular value required of $J_{F_\theta}$ by the regularization term, made the implicit manifold harder to optimize. This occasionally yielded plateaus in the loss function and resulted in incorrect manifolds, depicted in Figure 11. Training took 9.5 seconds.

The energy function for the EBIM consisted of 2 hidden layers with 32 units per hidden layer. It was trained for 10 epochs each wherein Langevin dynamics. We used a batch size of 100, a learning rate of 0.01, gradients clipped to a norm of 1, and energy magnitudes regularized with a coefficient of 1. Langevin dynamics at each training step was run for 10 steps with $\varepsilon = 0.1$, a step size of $\varepsilon^2$, and energy gradients clamped to maximum values of 0.03 at each step. Training took 1 minute, 8 seconds.

The pushforward EBM's encoder and decoder each had 3 hidden layers with 32 units per hidden layer. They were jointly trained for 500 epochs with a batch size of 50, a learning rate of 0.001, and gradients clipped to a norm of 1. Training took 33.5 seconds.

The pushforward EBM's energy function had 3 hidden layers and 32 units per hidden layer. It was trained for 50 epochs with a batch size of 100, a learning rate of 0.01, gradients clipped to a norm of 1, and energy magnitudes regularized with a coefficient of 0.1. Langevin dynamics at each training step were run for 60 steps with $\varepsilon = 0.5$, a step size of 10, and energy gradients clamped to maximum values of 0.03 at each step. Training took 1 minute, 53 seconds.

## B.2   Image Data

We parameterized the MDF in our EBM with a small UNet architecture modifed from the implementation in the labml.ai Python package (Jayasiri & Wijerathne, 2020), with layer widths scaled down by 75%. We chose a UNet because its skip connections give it full rank with a large output dimensionality ($28 \times 28 - 16 = 768$). All other image model architectures were based on the **Conv28** architecture of Yoon et al. (2021). All encoders and decoders were identical to that of **Conv28**. The constrained energy function for the EBIM models was also identical, except with an output dimension of 1. The energy functions for the AE+EBM, VAE+EBM, and EBIM, as well as the MDF, were trained with spectral normalization.

Nearly the same hyperparameters were used for both datasets, with the exception that a manifold dimension of 30 was used for the AE + EBM on Fashion MNIST (while 16 was used everywhere else).

The autoencoder in the AE + EBM was trained with a learning rate of $1 \times 10^{-4}$ and a batch size of 128 for 100 epochs. Gradients were clipped to a maximum norm of 10. The EBM in the AE + EBM was trained with a learning rate of $1 \times 10^{-5}$ and a batch size of 128 for 200 epochs. Gradients were also clipped to a maximum norm of 10. Energy norms were regularized with a coefficient of 0.1. During training, samples were initialize from the buffer 95% of the time. Langevin dynamics was run for 60 steps with a step size of 10, noise of 0.005, and energy gradients were clamped to maximum entries of 0.03.

The autoencoder in the VAE + EBM was trained using the same hyperparameters as in the AE + EBM, and the KL divergence term was trained with a weight of $1 \times 10^{-5}$. The energy was also trained with the same hyperparameters, except with a learning rate of $1 \times 10^{-4}$.

The hyperparameters used in Yoon et al. (2021) for the NAE were chosen for OOD detection; we changed them slightly for generative performance. For the latent on-manifold initialization step, we used a step size of 10 and a noise standard deviation of 0.005. In ambient space, we used noise with standard deviation 0.005. Models were pre-trained with the reconstruction loss alone for 100 epochs and then trained using the NAE loss for 100 more epochs.

The MDF in the EBIM was trained for 100 epochs with a learning rate of 0.0001 and batch size of 128. Empirically, we found it most effective to sample from and penalize negative samples for $\|F_\theta(x')\|^2$ while minimizing $\|F_\theta(x)\|$ (unsquared) for ground truth data. The negative sample square-norm magnitude was regularized with a coefficient of 0.1. We did not clip optimization gradients. Langevin dynamics gradients were clamped to 0.01 during, and we ran Langevin dynamics for 20 steps per training step. We regularized for a minimum and maximum singular value of 0.01, with a regularization coefficient of 100000. We sampled from the buffer during training with 95% probability.

The constrained EBM was trained for 100 epochs with a batch size of 64 and learning rate of 0.00001. Energy values were regularized with a coefficient of 1. Equation 14 was optimized with stochastic gradient descent with a learning rate of 1 for 60 steps per Langevin dynamics step, 1 of which was run per training step. Langevin dynamics steps were run with a step size 10, noise with a standard deviation of 0.005, and energy gradients clamped to a values of 0.005. We sampled from the buffer during training with 95% probability.

