# OpenReview forum: "Neural Implicit Manifold Learning for Topology-Aware Density Estimation"
_TMLR — Accepted by TMLR_

### Review · Reviewer_DBcR · 2023-09-07

**Summary Of Contributions:**

This paper presents a new topology-aware framework for constructing generative models when the data is supported on a lower-dimensional manifold. In this framework, the manifold is represented by the zero set of a neural network, named a neural implicit manifold, and a constrained energy-based model is introduced for density estimation on this manfiold. This method allows modelling more general manifolds, which is in contrast to prior works who define the manifold as the image of a smooth mapping on a low-dimensional latent space, thus can only capture manifolds that are almost diffeomorphic to a Euclidean space. Furthermore, the framework allows a straightforward representation of unions and intersections of neural implicit manifolds. Experiments on low-dimensional datasets with intriguing topologies and image datasets demonstrate the supreriority of the framework, in comparison with prior works, at capturing the topology of the data manifold, as well as generating images with better or comparable quality.

**Audience:**

Yes

**Broader Impact Concerns:**

I do not see any immediate implications which would require a Broader Impact Statement.

**Claims And Evidence:**

Yes

**Requested Changes:**

### Major Comments:
* The authors have mentioned that they believe a possible explanation for why NAE outperforms EBIM on Fashion MNIST is due to the computational complexity of the constrained Langevin algorithm. Is it possible to test this hypothesis by showing that with enough compute EBIM will outperform NAE on Fashion MNIST?

* As a recent popular generative model, denoising diffusion probabilistic models don't seem to fit in the framework of pushforward models. Is there a heuristic on why these models may or may not be able to properly capture the topology of the data in comparison with the proposed methods?

### Minor Comments:
* I am not an expert in this field, but at this point the experiments don't seem to be sufficient to prove the method's applicability in practical settings when data is high-dimensional. Having experiments on more complex image datasets such as CIFAR10 or stating that such experiments are not computationally tractable might help readers better understand the limitations of the current work.

**Strengths And Weaknesses:**

## Strenghts:
* The method is theoretically motivated and can provably model a wider class of manifolds than prior *pushforward models*.
* The framework makes it straightforward to model unions and intersections of neural implicit manifolds and generate samples from them.
* The experiments demonstrate that the framework is indeed successful at capturing the topology of the data manifold.

## Weaknesses:
* As the authors have also pointed out, the constrained Langevin algorithm can take a long time to converge and be inefficient in high dimensions.
* As a result of this inefficiency, the experiments are only run on relatively simple datasets.
* The method requires choosing the dimension of the data manifold *a priori*, and an unsuitable choice might lead to missing the topology of the manifold, which is the main advantage of this work in comparison with the literature.

---

> ### Author Response · Authors · 2023-09-20
> **Response to Reviewer DBcR**
>
> We thank you for the positive review. To address your questions:
> 1. You were curious whether the EBIM can outperform the NAE on Fashion MNIST with enough compute. In principle, we can test this by choosing as-close-as-possible parameter settings for the EBIM and NAE, and comparing the two regardless of runtimes. Unfortunately, this would be time-prohibitive for the EBIM: our EBIM uses 1 constrained Langevin dynamics step on the manifold per batch, while the NAE uses 10 in its on-manifold initialization phase. It is interesting though that the EBIM performs similarly to the NAE in spite of using much fewer Langevin dynamics steps in training. (We emphasize that constrained Langevin dynamics is only tractable at all because of the computational tricks provided in Section 3.2.)
> 2. You asked about whether diffusion models can capture manifolds of complex topologies. Please see our general response for some discussion on this.
> 3. You also discussed scaling the model up to high-dimensional data such as CIFAR-10. Informally, it is at least possible, but we do not have computational resources to train a full model and report results. In a preliminary experiment, we were able to fit the more-efficient first step of our model, the implicit manifold, unconditionally to CIFAR-10. It achieved a reasonable FID of 58.3 without the second step but it took over a week to train. Since we did not have the resources to fit and tune the second step, we will add only a brief mention of this to the final manuscript. We see further scaling this model as a promising direction for future research, and we note that our stochastic estimates and automatic differentiation tricks for scaling constrained Langevin Monte Carlo to over 700 dimensions is already a substantial improvement over what would naively be possible.

---

> > ### Comment · Reviewer_DBcR · 2023-10-10
> >
> > Thank you for your detailed responses! I'm happy with the current state of the manuscript and would like to recommend acceptance.

---

### Review · Reviewer_DZcr · 2023-09-15

**Summary Of Contributions:**

The authors of the paper propose a method to do generative modelling for
distributions supported on manifolds. They assume that the data lies on an
$m$ dimensional submanifold $\mathcal{M} \subset \mathbb{R}^n$ and learn the
distribution in a two-step procedure.

**Learning the Manifold**

The main novel idea is to characterize the manifold as the zero-set of a
function $F_{\theta}$,

$$ M_{\theta} = F_{\theta}^{- 1} (\{ 0 \}),$$

where $F_{\theta} : \mathbb{R}^n \rightarrow \mathbb{R}^{n - m}$. The
dimension $m$ of the manifold has to be known beforehand. An ad-hoc loss is
introduced to find a candidate $F_{\theta}$:

$$ \text{Loss} (\theta) =\mathbb{E} [\| F_{\theta} \| -
   \alpha \| F_{\theta} (x') \| + \beta (\eta - \| v^T J_{F_{\theta}} (x)
   \|)^2_+], $$
where $x \sim P^{\ast}, x' \sim \text{sg} [P_{\theta}], v \sim \text{Unif} (\text{Sphere})$.
To train such an $F_{\theta}$ one has to run MCMC to generate samples from
$P_{\theta} \propto \exp (- \| F_{\theta} \|^2)$ at each step and perform two
gradient operations through $F_{\theta}$ (once w.r.t. its inputs, and
then w.r.t. its parameters).

**Learning the Distribution on the Manifold**

After finding an optimal $\theta^{\ast}$, the distribution is learned as an
Energy-based, model, by performing gradient steps on

$$ \nabla_{\psi} p_{\psi} (x) = - \nabla_{\psi} E_{\psi} (x_i)
   +\mathbb{E}[\nabla_{\psi} E_{\psi}(x')], $$

where

$$ x' \sim P_{\psi } \propto \exp (- E_{\psi}) $$
is normalized to be a probability distribution on
$\mathcal{M}_{\theta^{\ast}}$.

To generate samples from $P_{\psi }$ for training or testing, one has to run
a constrained version of the Langevin algorithm. To that end, one has to solve
an optimization problem involving $F_{\theta}$ over an $n - m$ dimensional
space, to project the dynamics back onto the manifold. However, since this
optimization should be initialized close to its optimal value for small
step-sizes of the Langevin-algorithm, it should not take many iterations to converge.

For sampling, one has to run the constrained Langevin algorithm again.

**Audience:**

Yes

**Broader Impact Concerns:**

The reviewer does not have any ethical concerns concerning this work, except the ones that apply to any work on generative modeling or manifold detection.

**Claims And Evidence:**

No

**Requested Changes:**

See the section named "Weaknesses" in "Strengths and Weaknesses".

The main point for me to answer "No" for Claims and Evidence is $2.$. The statement "such methods are likely to specify all but the simplest of data topologies" is clearly not backed by evidence, as the most prominent example of push forward models, i.e. diffusion models and now flow models are both able to do so. The $f_\theta$ there is learned implicitly as the flow of an ODE.

Either discussing this point in more detail, showing how diffusion models indeed fail to learn simple data distributions, or removing statements of the kind would change my answer to that question.

**Strengths And Weaknesses:**

**Strengths**
- The idea of learning the manifold as a nullset of $F_{\theta}$ is
  attractive.
- The numerical experiments are convincing and seem to be the state of
  the art of energy-based models.


**Weaknesses**
1. The main drawback of the method is the large computational cost. One
  has to train two neural networks, both training procedures involving
  interleaving gradient steps with MCMC. Furthermore, one of the training procedures
  requires second-order derivatives of the neural network, while
  the other one requires intermediate steps between each of the MCMC steps to
  project the MC trajectory back to the manifold. This makes the comparisons
  to the other algorithms not informative. As detailed in the appendix, it
  seems like most of the other parameters (amount of training epochs, number
  of Langevin steps during training and generation) are chosen the same,
  which means that the samples generated by the other EBM-based models are
  computationally much cheaper to obtain. This should be at least clearly
  stated in the main text.
2. There should be a comparison to diffusion models for manifold data, as they are
  the state of the art for learning distributions as push-forward measures; it
  has been shown empirically as well as theoretically that diffusion models
  are able to sample manifold distributions. The authors should at least motivate
  why they think that their arguments about pushforward models do not apply here,
  or if they do. The reviewer thinks this could have to do with the fact that the drift of
  the reverse ODE in diffusion models will explode, making the optimal ODE flow $f_\theta$
  not continuous (and therefore not smooth).
3. It is unsatisfactory that even in the toy examples (von mises), where
  the authors had access to the true density/likelihood, they did not use it
  to compare model performance. An easy check would have been to compare
  the true likelihood on the training samples to the learned ones.
4. An experiment about out-of-distribution detection would have been very
  beneficial. The reviewer sees learning $F_{\theta}$ and using it to describe
  the manifold as the main contribution of the paper, and this would have been
  the most direct application of this part of the work.

---

> ### Author Response · Authors · 2023-09-20
> **Response to Reviewer DZcr**
>
> We thank you for your detailed and insightful review. Please see the below responses and let us know if you have any lingering concerns. We would be happy to discuss further.
> 1. You asked that we clearly state that samples from the EBIM are more expensive in the main text. We agree and have clarified this in the manuscript.
> 2. You expressed concerns about diffusion models. Since diffusion models are mathematically pushforward models, and yet they have been shown to adequately sample densities on manifolds, it would understandably seem that they violate our analysis about pushforward models. This is actually not the case, and the reasoning is subtle; please see our general response for clarification on this. Another subtle point, which we also clarify in the general response, is that diffusion models do not perform our target task, which is to learn densities *supported on manifolds*, and so are not a feasible point of comparison.
> 3. You mentioned that we did not compare models using likelihoods. We declined to do so because using likelihoods is actually frowned upon in the literature when estimating both manifolds and densities and is mathematically unprincipled, for a couple of reasons:
>      - Since all model manifolds will usually vary imperceptibly from the true manifold, any ground truth samples will fail to lie on the model manifold, meaning that the model will assign a likelihood of 0. Given that all density-on-manifold models will technically have likelihoods of zero for all test data points, a comparison between models would not be principled.
>      - In certain cases, ground truth samples can be projected to the manifold and their likelihoods can be approximated, alleviating the first concern. However, even then, when comparing models, every model will potentially learn a different manifold, meaning likelihoods are being evaluated on different manifolds. Mathematically, these are incomparable, and formally this is because the densities for each model are computed with respect to different base measures. Section 3A of Brehmer and Cranmer [A] (the arxiv version) is a good reference explaining how models can, for example, learn a distribution poorly while achieving infinitely high likelihoods by varying their manifolds.
>
>      Instead of likelihoods, we compared on the basis of Wasserstein-1 distances, which are generally considered the principled way to compare models when tractable in this context [B].
> 4. We did indeed test the ability of our implicit manifold trained on MNIST and Fashion MNIST to identify the other dataset as out-of-distribution (OOD); however, the Fashion MNIST model was unable to identify MNIST samples as OOD. It would appear that the same inductive biases causing likelihood-based models to rank MNIST and Fashion MNIST incorrectly [C] also affect the manifold learned by the EBIM. Given that our model exhibits the same pathological behaviour that affects likelihood-based models, would you still like us to evaluate EBIMs on other OOD-detection problems?
>
> [A] Brehmer, Johann, and Kyle Cranmer. "Flows for simultaneous manifold learning and density estimation." arXiv preprint arXiv:2003.13913 (2020).
>
> [B] Arjovsky, Martin, Soumith Chintala, and Léon Bottou. "Wasserstein generative adversarial networks." International conference on machine learning. PMLR, 2017.
>
> [C] Nalisnick, Eric, et al. "Do Deep Generative Models Know What They Don't Know?." International Conference on Learning Representations. 2018.

---

> > ### Comment · Reviewer_DZcr · 2023-09-20
> >
> > Thank you for the implemented changes and the honest answer to 4. Your findings seem quite interesting. I think including these findings into the paper (maybe elaborating a bit on what you wrote in point 4, potentially in the appendix), would be worthwhile.
> >
> > However, the already implemented changes alleviate my concerns and I recommend the paper for acceptance.

---

> > > ### Author Response · Authors · 2023-09-21
> > > **Thank you!**
> > >
> > > Thank you for the prompt response and recommending the paper for acceptance! We agree the OOD detection results are interesting and a bit surprising. We will include the discussion surrounding point 4 in the camera-ready version of the work.

---

### Review · Reviewer_uUEQ · 2023-09-17

**Summary Of Contributions:**

This paper develops an energy-based implicit manifold approach, which is a generative modeling framework for generating geometric data such as Geospatial data including global flood events, Amino acid modeling, as well as image generation including MNIST digits and Fashion MNIST.

The main idea of this approach is applying a stochastic gradient descent like method on the energy based model, which assigns a probabilistic density to each data point in the manifold. A key point here is the constrained projection step.

Then, experiments are conducted to validate the constrained langevin monte carlo algorithm on the defined manifold.

**Audience:**

Yes

**Broader Impact Concerns:**

This section is not applicable

**Claims And Evidence:**

No

**Requested Changes:**

- I find the "Method" section to be difficult to understand due to a lack of properly defined problem setup. Many of the derivations are not explained in enough detail to understand them step-by-step.
- The contribution of this paper is unclear to me because a) there is a lack of comparative studies in the experiments; b) the problem that Algorithm 1 is trying to solve is not clearly defined. Thus, a clear summary of contributions would be needed.
- The examples used in the experiments are fairly simple geometric objects. I wonder how would the new algorithm compare against (say) diffusion models or other more recent generative modeling (e.g., stable diffusion). This would be helpful for appreciating the significance of the results.

**Strengths And Weaknesses:**

S1. The paper includes a detailed discussion of the background, related work, and motivation, which is very helpful.

S2. The experiments are accompanied by figures illustrating the outcome of the generated images.

S3. Energy-based modeling is a promising direction in light of recent developments in generative modeling.

W1. In section 3, the description lacks a proper problem setup and problem formulation, thus, it is not completely clear what exactly is the problem the proposed algorithm is tackling exactly.

W2. Some of the steps in the algorithm are not completely justified. I'm listing a few here:

In section 3.1, condition 3 is stated as taking inspiration from Kumar et al. (2020). However, it is not clear to me where this condition is used later in the approach, or why it is needed here.

In Eq (7),  it is not clear to me where this loss is used later in the experiments. Some notations are not defined here, such as "sg," "P*." It is not clearly explained why U(S) (the uniform distribution) would be used here. Thus, I was not able to understand this loss.

The discussion concerning "Expressivity" does not follow naturally from the paragraph above; the logic is not clear either.

---

> ### Author Response · Authors · 2023-09-20
> **Response to Reviewer uUEQ**
>
> We sincerely thank you for the candid review. Most of your concerns were about clarity; in response, we have made numerous changes to the manuscript in blue and engaged with your concerns on a point-by-point basis. Outside of one request for clarification in the bullets below, this should be sufficient to address your concerns, but please let us know if you have any additional questions.
>
> Following your requested changes:
> - You found the “Method” section lacking in two respects:
>    - Problem setup: please see the **clarity** section below, where we clarify this and describe our changes to the manuscript.
>    - Derivations: please see the **specific technical concerns** section below, in the subsequent response.
> - You found the contributions unclear for two reasons:
>    - You cited a lack of comparative studies in the experiments. **Would you be able to define what you mean by “comparative study”?** We point out that all our experiments consist of quantitative and qualitative comparisons to relevant baselines, and the entire motivation of our work is intrinsically comparative (to pushforward models).
>    - You found the goal of Algorithm 1 to be stated unclearly: please see the **clarity** section below, where we clarify this and describe our manuscript changes.
>
>     We also point out that (1) the last paragraph of the introduction and (2) the first paragraph of the conclusion are both summaries of the paper’s contributions, with different points of emphasis.
> - You asked about potentially comparing our approach against more recent algorithms, like diffusion models. Please see our discussion about diffusion models in the general response.
>
>
> ### Clarity
> You had several concerns about clarity: primarily, you found it difficult to understand the problem setup and the purpose of the two main steps of our “Method” section, as embodied by Equation 7 [now Equation 8 after updates] and Algorithm 1, along with other parts of the methods section. While our problem setup and high-level outline of the method primarily live in our “Introduction” and “Background, Related Work, and Motivation” sections, we recognize that there is an opportunity to make the paper more accessible with better signposting in the “Method” section.
>
> We have added these “signposts” along with other changes directly addressing the specific concerns you brought up, along with a new section (3.3) summarizing our entire procedure. We believe these changes will be helpful in clarifying the problem setup and goal of these sections.
>
> To further clarify the flow of the methods section, we summarize it briefly here for your convenience. It focuses on two neural nets: an *implicit manifold*, which describes the data manifold, and a *constrained EBM*, which describes the underlying probability density within that manifold.
> 1. In section 3.1, we describe how to train the implicit manifold. The loss (now Equation 8) follows directly from the full-rank zero set theorem described in the background section.
> 2. In section 3.2, we describe the constrained EBM, whose training procedure follows from past work in the EBM literature, the constrained Hamiltonian Monte Carlo method of Brubaker et al. (2012), and our proposed method to efficiently carry out all the involved computations. The sampling procedure for this model, which we summarize in Algorithm 1, relies on the implicit manifold from 3.1. This sampling procedure is required for training as well, as is standard among EBMs.

---

> > ### Author Response · Authors · 2023-09-20
> > **Response to Reviewer uUEQ (continued)**
> >
> > ### Specific Technical Concerns
> > 1. You questioned the use of condition (3) in section 3.1, saying we did not justify it. We have reworded a couple of sentences in section 3.1 to make the below justification more explicit.
> >
> >
> >    To clarify, condition (3) follows directly from the full-rank zero set theorem we discuss in the background section. Indeed, the 3 conditions in section 3.1 are just a rewording of the theorem.
> >
> >
> >    To further clarify, it is not condition (3) per se that takes inspiration from Kumar et al. (2020), but rather our *means of enforcing it*.
> > 2. You asked where our loss from Equation (7) [now Equation 8 after our updates] is used in the experiments. This is the loss used to train the implicit manifold $F_\theta$ discussed throughout the paper.
> > 3. You also asked about the meaning of some notation in the methods section. For the methods, we carried forward notation defined in the previous sections: sg is the stop-gradient operator (defined in section 2.3) and $P^*$ is the ground-truth distribution we are trying to estimate (defined in section 1).
> > 4. You also asked why we used the uniform distribution over the unit sphere for sampling here. Since we want to bound $||v^T J_{F_\theta}(x)||$ away from zero for all unit-vectors $v$, we sample uniformly random unit-vectors $v$ and bound the corresponding term away from zero, so that at the optimality of this term, all unit vectors will be bounded away from 0. This is the inspiration we mentioned from Kumar et al. (2020); hopefully the correction from above makes this clearer. We added some further clarification to bridge this part of the loss to the discussion in the paper.
> > 5. You said the logic of the “expressivity” section is not clear. No specific math is actually developed here. It consists primarily of two claims:
> >     -  Pushforward models can describe densities on precisely the data manifolds that are diffeomorphic to a subset of Euclidean space. This claim is discussed at length in the “Density Estimation with Pushforward Models” and “Topological Challenges” subsections of the background, so we do not further elaborate on it here.
> >     - Implicit manifolds (our model) can describe data manifolds satisfying a specific technical condition (that the manifold’s normal bundle is trivial). We do not elaborate on this claim, as it is beyond the scope of our paper, and instead point to Lee (2013) as a reference. While we are unable to provide a simple characterization of exactly what this technical condition implies, it is known to allow for a larger set of manifold geometries than the condition for pushforwards.
> >
> >     We slightly modified one of the sentences to connect the first claim to the background section.
> > 7. You said “the problem Algorithm 1 is trying to solve is not clearly defined.” Algorithm 1 is our efficient implementation of Constrained Langevin Monte Carlo (CLMC), the procedure used to sample from $P_{\theta^*, \psi}$, the probability distribution of our constrained EBM. We point out that the goal of CLMC is built up carefully over the 4 paragraphs from the start of Section 3.2 up to the first paragraph under the “Constrained Langevin Monte Carlo” paragraph heading.

---

> > > ### Author Response · Authors · 2023-11-08
> > > **Any remaining concerns?**
> > >
> > > Dear reviewer **uUEQ**,
> > >
> > > We just wanted to check in again regarding the point-for-point changes and clarifications we've made in response to your list. Did you have any further concerns?

---

### Author Response · Authors · 2023-09-20
**General Response**

We thank all reviewers for the considerable time and effort they have spent reviewing our paper. We look forward to a productive discussion period and hope all concerns can be addressed. In response to your concerns, and as described in the individual responses, we have made a number of updates to the manuscript. These updates are coloured blue for your convenience.

All reviewers asked about the contrast between our models and diffusion models. Reviewer **DBcR** asked whether diffusion models can be used for capturing data topologies. Reviewers **DZcr** and **uUEQ** wondered how diffusion models would perform on our experiments, and reviewer **DZcr** asked about the relationship between diffusion models and pushforward models.

These questions are natural, as diffusion models are popular and have been shown to be able to fit distributions on manifolds with potentially arbitrary topologies [A]. *The key point to highlight, however, is that the purpose of our method is to estimate **probability densities on manifolds**, and diffusion models cannot do this.*

1. At first, this statement might seem strange, as diffusion models are known to be able to learn manifolds [A] and to provide densities [B]. However, they cannot do both at once:
     - As reviewer **DZcr** speculated, when a diffusion model learns a manifold, the drift term of the reverse ODE approaches infinity as the time approaches 0 [A]. This makes the diffusion model’s ODE solution neither smooth nor invertible.
     - On the other hand, the validity of density estimates is predicated on the invertibility of the ODE (ie. it must represent a normalizing flow) [A].
     As a result, diffusion models meet a similar fate to pushforward models as described in the second last paragraph of page 3 of the manuscript: when they model manifolds, their bi-Lipschitz constants go to infinity, making their density estimates unreliable.
2. Mathematically speaking, reviewer **DZcr** is correct that, when their solutions are invertible, diffusion model ODEs result in continuous normalizing flows, which are pushforward measures through an invertible function $f_\theta: \mathbb{R}^n \to \mathbb{R}^n$. However, in our work we define pushforward models for density estimation on manifolds as maps from $\mathbb{R}^m$ to $\mathbb{R}^n$ with $m < n$. The reason for this is that full-dimensional maps provide full-dimensional densities - these cannot necessarily be related in any sense to densities on manifolds. Formally, these are densities taken with respect to the Lebesgue measure, while implicit manifolds and pushforward models have densities with respect to their respective Riemannian measure. Thus, even when numerically stable, diffusion models do not provide densities on manifolds.

Since diffusion models do not provide densities on manifolds, we cannot compare them to EBIMs. They also lack a direct characterization of the manifold, whereas our model produces a clean mathematical characterization as a null set. (These same two things are true of continuous normalizing flows.) Furthermore, as state-of-the-art image synthesizers, diffusion models can surely surpass EBIMs on MNIST and Fashion MNIST in terms of generation quality. For all of these reasons, we believe that comparing against diffusion models would be uninformative. We thank the reviewers for raising these important discussion points; we have modified the background and conclusion sections accordingly.

[A] Pidstrigach, Jakiw. "Score-based generative models detect manifolds." Advances in Neural Information Processing Systems 35 (2022): 35852-35865.

[B] Song, Yang, et al. "Score-Based Generative Modeling through Stochastic Differential Equations." International Conference on Learning Representations. 2020.

---

### Author Response · Authors · 2023-09-28
**Final Decisions Open Soon**

We kindly ask our reviewers: do you have any additional concerns or requested changes before the recommended 2-week discussion period wraps up on Saturday?

In particular, reviewer **uUEQ** had several concerns about clarity, in response to which we carefully reworded a number of passages in the manuscript, added a new method summary section, and provided a list of explanations via direct response. Were these changes satisfactory?

---

### Author Response · Authors · 2023-12-21
**Camera-Ready Upload Complete**

The camera-ready version of the manuscript has now been uploaded to OpenReview. Thank you to all who participated in the review process.

---

### Decision · Action_Editor_2vK7 · 2023-11-30

**Recommendation:** Accept as is

**Comment:**

I think the paper is fine as it is. It does not oversell its contributions, the full rank zero set theorem is interesting, and its implementation through (8) is also a nice read. The overall algorithm is sound, though has limitations (slowness of Langevin for that setup) as outlined by reviewers.

**Audience:**

TMLR readers interested in EBMs, with a keen interest on geometric / constrained data.

**Claims And Evidence:**

The paper proposes to learn a density on a manifold parameterization of data samples using a neural implicit model. The overall approach is two step: learn the manifold as the set of roots of a NN by crafting a loss that considers maps that satisfy the "full rank zero set theorem" at all those roots, which results in objective (8), and then fine tuning / learning a density on that manifold using constrained Langevin (10).

The paper is interesting, the only major limitation is the relative simplicity of experiments, which consider mostly R^3 or black and white images (e.g. MNIST). The approach is overall correct and well motivated.